METHODS

# Optimal transport reveals dynamic gene regulatory networks via gene velocity estimation

**Wenjun Zhao** [1,2*], **Erica Larschan**[3], **Björn Sandstede** [1], **Ritambhara Singh**[4]

**1** Division of Applied Mathematics, Brown University, Providence, Rhode Island, United States of America, **2** Department of Mathematics, University of British Columbia, Vancouver, Canada, **3** Department of Molecular Biology, Cell Biology and Biochemistry, Center for Computational Molecular Biology, Brown University, Providence, Rhode Island, United States of America, **4** Department of Computer Science, Center for Computational Molecular Biology, Brown University, Providence, Rhode Island, United States of America

* zhaow@wfu.edu

**Data availability statement:** Data availability statement Datasets simulated by HARISSA are

## Abstract

Inferring gene regulatory networks from gene expression data is an important and challenging problem in the biology community. We propose OTVelo, a methodology that takes time-stamped single-cell gene expression data as input and predicts gene regulation across two time points. It is known that the rate of change of gene expression, which we will refer to as gene velocity, provides crucial information that enhances such inference; however, this information is not always available due to the limitations in sequencing depth. Our algorithm overcomes this limitation by estimating gene velocities using optimal transport. We then infer gene regulation using time-lagged correlation and Granger causality via regularized linear regression. Instead of providing an aggregated network across all time points, our method uncovers the underlying dynamical mechanism across time points. We validate our algorithm on 13 simulated datasets with both synthetic and curated networks and demonstrate its efficacy on 9 experimental data sets.

## Author summary

Understanding how genes interact to regulate cellular functions is crucial for advancing our knowledge of biology and disease. We present OTVelo, a method that uses single-cell gene expression data collected at different time points to infer gene regulatory networks. By leveraging the mathematical framework of optimal transport, OTVelo first reconstructs likely trajectories of gene expression changing over time, then infers the regulatory interactions from the rate of change along these trajectories. This approach allows OTVelo to offer a dynamic view of how gene interactions change over time, providing deeper insights into cellular processes. Unlike traditional methods, which often rely on static network assumptions, OTVelo captures temporal information through

available from the original publication of CARDAMOM [12] (https://github.com/eliasventre/cardamom/tree/main/results_article/Benchmark_on_simulated_data), and curated models with pseudotime and all dropout rates are available along with BEELINE [16] (https://zenodo.org/records/3701939). The standard benchmarks for scRNA-seq, including the expression data, pseudotime, highly variable genes, list of transcription factors, and ground truth networks, were also provided by BEELINE [16]. The scGEM dataset is provided by the supplementary material of Cheow et al. [34], and THP-1 differentiation dataset can be found from SINCERITIES [7] (https://github.com/CABSEL/SINCERITIES). For mouse data, we used the raw count matrices provided by Semrau et al. [36] (https://www.ncbi.nlm.nih.gov/geo/query/acc.cgi?acc=GSE79578). The *drosophila* data can be found in Calderon et al. [37] (https://www.ncbi.nlm.nih.gov/geo/query/acc.cgi?acc=GSE190149). The implementation of our algorithm OTVelo, as well as results that reproduce the figures, can be found on https://github.com/sandstede-lab/OT-Velocity.

**Funding:** BS was partially supported by the NSF under grants DMS-2038039 and DMS-2106566. RS's contribution was funded by NIH award 1R35HG011939-01. The funders had no role in study design, data collection and analysis, decision to publish, or preparation of the manuscript.

**Competing interests:** The authors have declared that no competing interests exist.

ancestor-descendant transitions without assuming a specific underlying regulatory model. We validate our approach using both simulated and real-world data, demonstrating its effectiveness in revealing complex gene regulation patterns. This method could lead to new discoveries in understanding biological systems and developing disease treatments.

## Introduction

Gene regulatory networks (GRNs) have been used to describe the behavior in biological processes and provide crucial understanding of complex mechanisms, such as cellular development and their response to stimuli. Inference of such networks involves (1) identifying the upstream and downstream genes to determine the direction of the regulatory relation, and (2) quantifying the type and strength of the interaction. Traditional approaches were based on observations of cell populations, where information might be averaged out when aggregating information from a heterogeneous population cells. Single-cell RNA sequencing (scRNA-seq) [1] enabled observations of individual cells and powered inference methods with larger sample sizes. Knowing how gene expression changes over time is crucial to understanding cell development and gene regulation. Recent advances of scRNA-seq technologies allow experimentalists to collect single-cell gene expression data from cell populations at multiple time points. Alternatively, pseudotimes [2] can be inferred from single-cell data by computationally ordering cells along cell trajectories in situations where temporal information is not directly available

During the past decades, many algorithms have been developed that use single-cell RNA-seq data to infer GRNs. For example, GENIE3 [3] and its variant SCENIC [4] infer regulation by a tree-based method for predicting the gene expression levels, while other methods explore time-lagged correlation [5] or information metrics [6]. Another popular family of model-free methods, including SINCERITIES [7] and SCRIBE [8], uses Granger causality to determine causal relations by testing (1) whether the cause happened before the outcome and (2) whether the cause provides unique information for predicting the outcome. To integrate temporal information into the pipeline, several model-based methods were developed that assume that gene expression data obey certain ODE models. SCODE [9] and GRIT [10] both use a linear constant-coefficient model to predict an aggregate GRN by fitting the model to observations: SCODE utilizes linear regression, while GRIT relies on optimal transport. HARISSA [11] and CARDAMOM [12] assume that expression data follow a reaction model whose parameters are estimated using coarse-graining and regression. Finally, scEGOT [13] approximates cell populations using Gaussian mixture models and utilizes optimal-transport of the mixture variables to estimate GRNs based on a linear ODE model. Other recent approaches use stochastic differential equations (SDE) as a reference for optimal transport: [14] refines coupling through iterative updates with respect to an SDE, and [15] applies a similar approach to knockout data. However, as suggested by recent benchmarking work [16], the performance of existing algorithms is only slightly better than random baselines, and the algorithms are generally better at recovering synthetic networks than recovering experimental datasets or Boolean models of curated biological networks. Moreover, algorithms that do not require temporal information such as [3] and [6] appear to be more accurate than methods that require pseudotime ordering, which suggests that the temporal information is more challenging to utilize for GRN inference.

The identifiability of GRNs would be greatly improved if cell trajectories could be obtained at single-cell resolution [8]. However, because scRNA-seq destroys cells in the course of

recording their gene expression levels, it is impossible to track gene counts in the same cell across time. It has been shown that the computationally inferred pseudotimes can fail to capture the true temporal evolution [8,17]. An alternative approach is to incorporate RNA-velocity [18,19] into the algorithm, which helps to restore some of this information [8,17]. RNA velocity is an estimate of the rate of change of gene expression levels based on the splicing dynamics, which provides temporal information within each cell and has shown its potential for GRN inference tasks in methods such as Velorama [20]. Although proven helpful, RNA velocity cannot be used to calculate the velocity of gene expression counts without measuring a sufficient amount of spliced and unspliced RNA.

We propose OTVelo to infer dynamic gene-regulatory networks from estimates of gene velocities (that is, the rate of change of gene expression levels) obtained from optimal transport (see Fig 1). Our approach consists of two steps that take time-stamped gene count matrices as input. In the first step, we model the ancestor-descendant transition via optimal transport (OT) between consecutive time points, following the setup in Waddington-OT [21]. Optimal transport allows us to predict the gene expression level of each cell at past and future times. We utilize this information to estimate the rate of change of gene expression levels separately for each gene within each cell using finite differences: we refer to this estimate as the gene velocity and note that we achieve single-cell resolution for each gene and that this approach does not rely on the measurement of splicing dynamics. In the second step, we infer GRNs across time points by employing (1) a generalized version of time-lagged correlation or, alternatively, (2) Granger causality, adapted to use transition probabilities rather than one-to-one correspondences as seen in real trajectories. We note that, in contrast to the optimal-transport based approaches we reviewed above, our algorithm is non-parametric and does not rely on an underlying model of gene expression and regulation. We validate our algorithm on (1) nine synthetic networks derived from simulations with HARISSA [11] and (2) four curated networks adapted from known biological processes based on Boolean models provided by [16]. We also apply our algorithm to four experimental datasets from mouse, human, and *drosophila* that provide both qualitative and quantitative validation. Our method consistently achieves very good performance, ranking either as the best or amongst the best compared to several other algorithms across almost all datasets, showing that OT-based methods have great potential for future exploration of GRN inference.

## Methods

Modern single-cell sequencing technology enables observations at single-cell resolution. However, given the destructive nature of this technique, we can measure gene expression levels only once, at a given time point, for each cell. To infer how genes are regulated over time, we therefore need to predict the temporal trajectories of each cell from the observed data, which consist of gene expression counts of different cell populations at different time points. Once we identify temporal trajectories of each cell, we can attempt to infer gene regulatory pathways by analysing how gene velocities (that is, the rate of change of gene expression levels over time) are aligned in time across genes within each cell. Our proposed pipeline, summarized in Fig 1, accomplishes these two tasks by predicting cell trajectories using optimal transport (OT) and inferring gene regulation using time-lagged correlation or, alternatively, Granger causality applied to the gene velocities we calculated from the predicted cell trajectories. In this section, we provide the mathematical details of the proposed algorithm.

The input to our pipeline consists of time-stamped single-cell gene expression data that are observed at $N$ distinct time points. At each time $t$, a sample of $n(t)$ cells is drawn, and the RNA reads of $m$ genes are used to form an $m \times n(t)$ count matrix at time $t$. In particular,

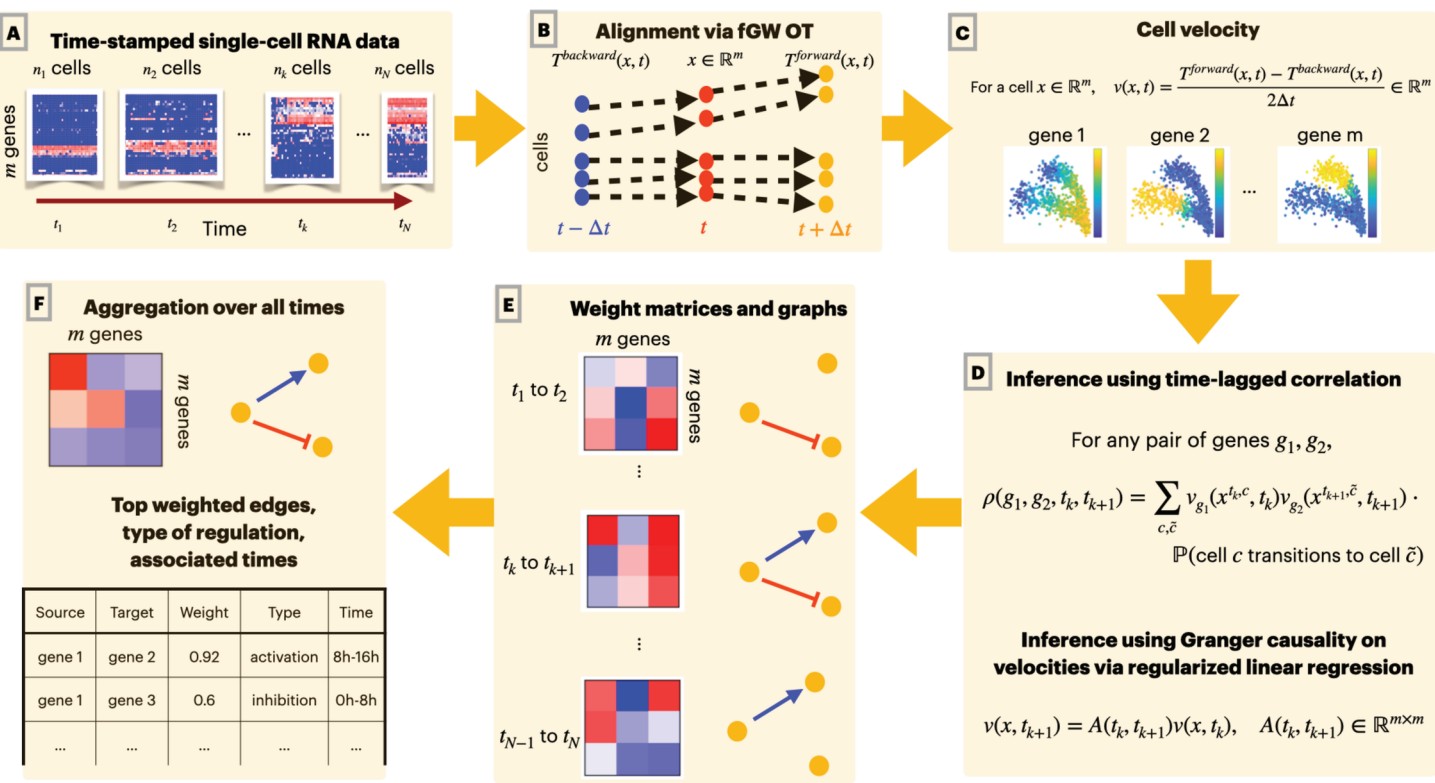

**Fig 1. Overview of our pipeline:** (A) The input of our algorithm are time-stamped single-cell gene expression data. At each time point, a different cell population is sampled and their gene expression level measured. (B) We use optimal transport (OT) to predict past and future states (represented by the expression levels of all genes) of each cell. (C) For each cell, we approximate the gene velocity (the rate of change of expression levels) of each gene using the finite difference of predicted past and future states. (D) To infer gene regulation, we use either time-lagged correlation analysis or Granger causality of gene velocities separately for each pair of adjacent time points. (E) Since the analysis is run independently for each time interval, we obtain temporal GRNs, which are represented by graphs that evolve over time. (F) These graphs can be consolidated into a single global gene regulatory network.

each column in the count matrix corresponds to an individual cell, and each entry records the number of RNA reads mapped to a specific gene in that cell. Our goal is to identify the regulatory relation between these $m$ genes. We summarize the notation we will use throughout this work in Table 1.

## Fused Gromov–Wasserstein optimal transport predicts cell trajectories

Given count matrices collected at two consecutive time points $t$ and $\tilde{t}$, optimal transport seeks to identify a transition matrix that measures the probability likelihood that a cell at the second time point $\tilde{t}$ is the descendant of a cell observed at time $t$. The transition matrices are computed as solutions to an appropriate optimal transport problem in which we minimize an overall cost function. In particular, in contrast to methods that utilize dynamic optimal-transport (OT) formulations or deep learning frameworks to identify continuous trajectories, such as MIOFlow [22] and TIGON [23], we follow the original approach from Waddington-OT [21] to find transition matrices as solutions to optimal transport problems. However, instead of using classical optimal-transport cost functions as in [21], we employ fused Gromov–Wasserstein optimal transport [24]: Gromov–Wasserstein optimal transport accounts for local geometry present in the underlying data and was previously used, for instance, in [25] to align multiomics data. We first state the optimal transport problem

**Table 1. Notation of quantities used in the description of the proposed methodology.**

| Notation | Description |
|---|---|
| $g \in \{1, ..., m\}$ | index for genes |
| $t \in \{t_1, ..., t_N\}$ | index for time points |
| $c \in \{1, ..., n(t)\}$ | index of each of the $n(t)$ cells observed at time $t$ |
| $x_g^{t,c} \in \mathbb{R}$ | expression level of gene $g$ for cell $c$ observed at time $t$ |
| $x^{t,c} \in \mathbb{R}^m$ | expression level of all genes for cell $c$ observed at time $t$ |
| $x^t \in \mathbb{R}^{m \times n(t)}$ | count matrix for the cells observed at time $t$ |
| $v_g(x^{t,c}, t) \in \mathbb{R}$ | velocity of gene $g$ in cell $c$ at time $t$ |

we consider here and then explain the various variables and spaces that are involved in this formulation. The desired transition matrix is found as

$$T^{t,\tilde{t}} := \underset{T \in \Pi(p,q) \subset \mathbb{R}^{n(t) \times n(\tilde{t})}}{\arg\min} \left[ (1-\alpha)\langle T, D^{t,\tilde{t}} \rangle_F + \alpha \sum_{\substack{1 \le c,d \le n(t), \\ 1 \le \tilde{c},\tilde{d} \le n(\tilde{t})}} L_{\text{GW}}(S_{c,d}^t, S_{\tilde{c},\tilde{d}}^{\tilde{t}}) T_{c,\tilde{c}} T_{d,\tilde{d}} \right], \quad (1)$$

where:

- $D^{t,\tilde{t}} \in \mathbb{R}^{n(t) \times n(\tilde{t})}$ captures differences between cells across time points, where each entry $D_{c,\tilde{c}} = d(x^{t,c}, x^{\tilde{t},\tilde{c}})^2$ measures the difference between cell $c$ sampled at time $t$ and cell $\tilde{c}$ sampled at time $\tilde{t}$ (e.g., $d(x, \tilde{x}) = \|x - \tilde{x}\|$ is the Euclidean distance in $\mathbb{R}^m$);
- $S^t \in \mathbb{R}^{n(t) \times n(t)}$ with entries $S_{c,d}^t = d_S(x^{t,c}, x^{t,d})$ captures the difference between cells sampled at the same time $t$, where $d_S$ is a metric on the collection of gene-expression vectors (see Preprocessing section for our specific choice);
- $L_{\text{GW}}: \mathbb{R}^2 \to \mathbb{R}^+$ accounts for the misfit of pairwise distances in the two spaces (e.g., $L_{\text{GW}}(r_1, r_2) = |r_1 - r_2|$);
- $\langle A, B \rangle_F = \text{Tr}(A^T B) = \sum_{i,j} A_{ij} B_{ij}$ for $A, B \in \mathbb{R}^{l \times \tilde{l}}$;
- $\alpha$ is a trade-off parameter that controls the ratio of the Wasserstein and Gromov–Wasserstein loss terms;
- $\Pi(p,q) = \{T \in \mathbb{R}^{n(t) \times n(\tilde{t})} : T\mathbf{1} = p, \ T^T\mathbf{1} = q, \ T \ge 0\}$, where $p$ and $q$ are marginal densities specifying the weight of each cell. The default value is a uniform distribution where $p_c = \frac{1}{n(t)}$ for any $c \in \{1, ..., n(t)\}$, and $q_{\tilde{c}} = \frac{1}{n(\tilde{t})}$ for any $\tilde{c} \in \{1, ..., n(\tilde{t})\}$.

As mentioned above, the optimal coupling matrix $T^{t,\tilde{t}}$ measures the likelihood that a cell at time $\tilde{t}$ descended from a cell at time $t$. The fused Gromov–Wasserstein problem is a generalization of OT that considers both the similarity in original feature space (through the Wasserstein loss) and the global structure (through the Gromov–Wasserstein loss) [24,26]. Both features are crucial for time-series data, as it would be preferable to enforce continuity in feature space and capture global behavior such as branching into different cell types. Empirically, for better computational efficiency, the problem is solved via its entropic-regularized version [26] using projected gradient descent:

$$T^{t,\tilde{t}} := \underset{T \in \Pi(p,q) \subset \mathbb{R}^{n(t) \times n(\tilde{t})}}{\arg\min} \left[ (1-\alpha)\langle T, D^{t,\tilde{t}} \rangle_F + \alpha \sum_{\substack{1 \le c,d \le n(t), \\ 1 \le \tilde{c},\tilde{d} \le n(\tilde{t})}} L_{\text{GW}}(S_{c,d}^t, S_{\tilde{c},\tilde{d}}^{\tilde{t}}) T_{c,\tilde{c}} T_{d,\tilde{d}} - \epsilon H(T) \right], \quad (2)$$

where $H(T) = \sum_{c,\tilde{c}} T_{c,\tilde{c}} \log(T_{c,\tilde{c}})$ is the entropic regularization term, and $\epsilon > 0$ is a parameter that controls the dispersion of the coupling matrix $T$. Larger values of $\epsilon$ will converge faster but create more dispersion in $T$, while smaller $\epsilon$ gives a more precise matching but will typically converge slower.

As in [21], we can use the coupling matrices $T$ found as solutions to the fused Gromov–Wasserstein problem (2) to predict cell states across different time points. Assume that $n(t)$ cells are observed at time $t$, $n(\tilde{t})$ cells are observed at time $\tilde{t}$, and the coupling matrix $T^{t,\tilde{t}}$ satisfies (2). For each individual cell $c$ observed at time $t$ via its expression vector $x^{t,c}$, we can then infer its descendent at time $\tilde{t}$ via the barycentric projection

$$\mathcal{T}^{t,\tilde{t}}(x^{t,c}) = \sum_{\tilde{c}} \frac{T_{c,\tilde{c}}^{t,\tilde{t}}}{\sum_{\tilde{d}} T_{c,\tilde{d}}^{t,\tilde{t}}} x^{\tilde{t},\tilde{c}}, \tag{3}$$

which is a weighted average of the expression vectors $x^{\tilde{t},\tilde{c}}$ of the cells $\tilde{c}$ observed at time $\tilde{t}$ with weights according to the probability that they arise from cell $c$ at time $t$.

## Temporal cell trajectories predict gene velocities

The barycentric projection map $\mathcal{T}^{t_k,t_{k+1}}$ defined in (3) represents the dynamics for a cell moving from time $t_k$ to $\tilde{t}_{k+1}$ in the gene expression space, and the associated gene velocity (the rate of change of expression levels) forward in time can be defined as:

$$v^{\text{forward}}(x, t_k) = \frac{\mathcal{T}^{t_k,t_{k+1}}(x) - x}{t_{k+1} - t_k}.$$

The gene velocity represents the direction and speed of change within a cell, and it is defined for each gene. If the velocity of a gene in a cell is positive, this suggests that the gene is being activated in that cell, whereas a negative gene velocity suggests repression. We can similarly define the backward gene velocity:

$$v^{\text{backward}}(x, t_k) = \frac{x - \mathcal{T}^{t_k,t_{k-1}}(x)}{t_k - t_{k-1}}.$$

To include both forward and backward information, we use, whenever possible, the following centered finite-difference expression to predict gene velocity:

$$v^{\text{center}}(x, t_k) = \frac{t_k - t_{k-1}}{t_{k+1} - t_{k-1}} v^{\text{forward}}(x, t_k) + \frac{t_{k+1} - t_k}{t_{k+1} - t_{k-1}} v^{\text{backward}}(x, t_k).$$

With equally spaced time points where $t_k - t_{k-1} = t_{k+1} - t_k$, the centered finite-difference expression reduces to:

$$v^{\text{center}}(x, t_k) = \frac{v^{\text{forward}}(x, t_k) + v^{\text{backward}}(x, t_k)}{2}.$$

We use forward velocity for the first time point, centered velocities for all intermediate time points, and backward velocity for the last time point:

$$v(x, t_1) = v^{\text{forward}}(x, t_1), \quad v(x, t_k) = v^{\text{center}}(x, t_k) \text{ for } k = 2, ..., N-1, \quad v(x, t_N) = v^{\text{backward}}(x, t_N). \tag{4}$$

Note that the expression for $v(x, t_k)$ above applies only to cells $x^{t_k}$ that are observed at time $t_k$. To generalize it to cells observed at other time points $\tilde{t} = t_j$ with $j \neq k$, we use the barycentric projection

$$\hat{v}(x^{\tilde{t},\tilde{c}}, t) := \sum_{c=1}^{n(t)} \frac{T_{c,\tilde{c}}^{t,\tilde{t}}}{\sum_d T_{d,\tilde{c}}^{t,\tilde{t}}} v(x^{t,c}, t), \tag{5}$$

which is a weighted average of $v(x^t, t)$.

**OT velocity and RNA velocity.** The idea behind the OT-based gene velocities we defined in (4) is to estimate the rate of change for each cell similar to what RNA velocities [18,19] achieve, but without using splicing dynamics, which may not be available due to insufficient depth of sequencing. As a comparison, we calculate cell velocity fields using OT velocity and RNA velocity respectively, for a mouse pancreas data [27] and project them into UMAP spaces. Our results presented in Fig 2 show qualitative agreement even though OT velocity does not utilize the spliced and unspliced information.

## Time-lagged correlation predicts temporal relation of gene velocities

To quantify the temporal order of changes of gene expression levels across genes, we define the OT-based time-lagged correlation between any two genes $g_1$ and $g_2$ by the expression

$$C_{g_1,g_2} := \frac{1}{N-1} \sum_{k=1}^{N-1} \sum_{c=1}^{n(t_k)} \sum_{\tilde{c}=1}^{n(t_{k+1})} v_{g_1}(x^{t_k,c}, t_k) v_{g_2}(x^{t_{k+1},\tilde{c}}, t_{k+1}) T_{c,\tilde{c}}^{t_k,t_{k+1}}, \tag{6}$$

where $v_g(x^{t,c}, t)$ represents the velocity of gene $g$ in cell $c$ at time $t$. The product of the gene velocities is weighted by the likelihood that cell $\tilde{c}$ at time $t_{k+1}$ is the descendent of cell $c$ at time $t_k$, which is measured by the coupling matrix $T^{t_k,t_{k+1}}$, and summed up across all time points. Note that the expression (6) can be interpreted as a generalization of the canonical time-lagged correlation, which is defined for the temporal trajectory of a single cell (so $T \equiv 1$). To

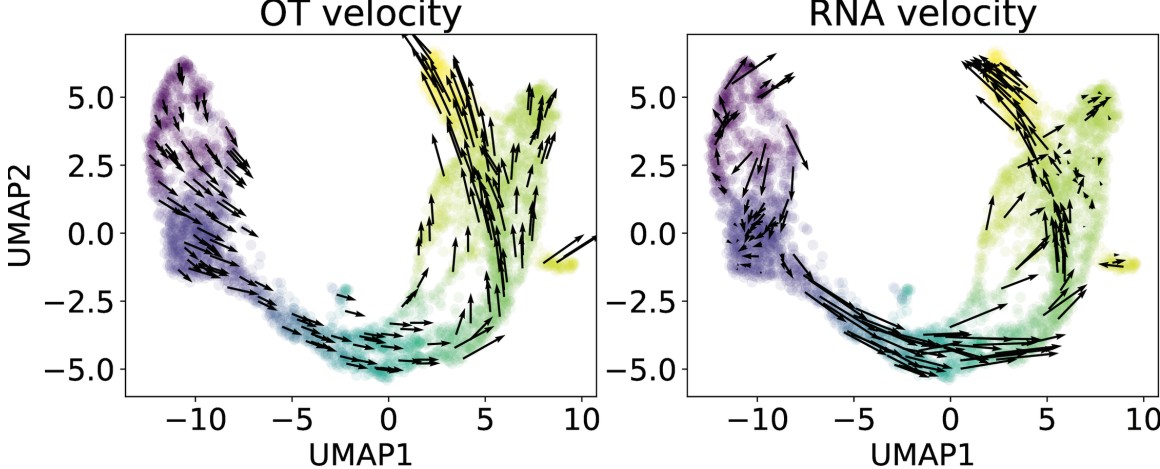

**Fig 2. Comparison between optimal transport-based cell velocities (left) and RNA velocities (right) computed from scVelo [18] and projected onto 2D UMAP coordinates.** To create data points for optimal transport, we use pseudotime computed from scVelo (indicated by color) and divide it uniformly according to quantiles.

account for the fact that different genes may have different magnitudes of their expression levels, we normalize the velocities for different genes to have unit standard deviation across all cells at all time points.

The sign and amplitude of the time-lagged correlation across different genes can reveal crucial information about gene interaction. If $C_{g_1,g_2} \gg 1$, this suggests that $g_1$ and $g_2$ are regulated in the same direction and that the regulation of $g_1$ happens before $g_2$: this suggests a possible activation of gene $g_2$ by gene $g_1$. In contrast, if $C_{g_1,g_2} < 0$, this suggest an inhibition of $g_2$ by $g_1$.

**Remark 1 (unit-free):**  The matrix $C$ is unit-free, since all velocities are normalized to unit standard deviation and multiplied by a non-dimensional probability coupling matrix whose entries add up to one.

**Remark 2 (generalization to different time lags):**  The time-lagged correlation in (6) applies to consecutive time points $(t_k, t_{k+1})$. This expression can be generalized to different time gaps lag > 0 via

$$C_{g_1,g_2}^{\mathrm{lag}} := \frac{1}{N-\mathrm{lag}} \sum_{k=1}^{N-\mathrm{lag}} \sum_{c=1}^{n(t_k)} \sum_{\tilde{c}=1}^{n(t_{k+\mathrm{lag}})} v_{g_1}\left(x^{t_k,c}, t_k\right) v_{g_2}\left(x^{t_{k+\mathrm{lag}},\tilde{c}}, t_{k+\mathrm{lag}}\right) T_{c,\tilde{c}}^{t_k, t_{k+\mathrm{lag}}},$$

where the coupling $T^{t_k, t_{k+\mathrm{lag}}}$ is computed via the composition of 'lag' consecutive coupling matrices.

## Granger causality predicts temporal relation of gene velocities

The correlation matrix $C$ defined in (6) is often dense, as many genes can be highly correlated. To enhance the sparsity of predicted edges and probe for causality, we can alternatively adopt the notion of Granger causality [28] which assesses whether (1) the cause happens prior to its effect and (2) the cause has unique information about the future values of its effect. We follow the approach described in SINCERITIES [7] through a regularized linear regression step between any pair of consecutive time points $t_k$ and $t_{k+1}$:

$$A^{t_k, t_{k+1}} = \underset{A \in \mathbb{R}^{m \times m}}{\arg\min} \left[ \left\| v(x^{t_{k+1}}, t_{k+1}) - A\hat{v}(x^{t_{k+1}}, t_k) \right\|_{\mathbb{R}^{m \times n(t_{k+1})}} + \lambda \left( r\|A\|_1 + (1-r)\|A\|_2 \right) \right], \quad (7)$$

where:

- $v(x^t, t) = \left(v(x^{t,c}, t)\right)_{c=1,\dots,n(t)} \in \mathbb{R}^{m \times n(t)}$ is the vector of the cell velocities $v(x^{t,c}, t)$ defined in (4), and similarly for $\hat{v}(x^t, \tilde{t}) = \left(\hat{v}(x^{t,c}, \tilde{t})\right)_{c=1,\dots,n(t)} \in \mathbb{R}^{m \times n(t)}$ with $\hat{v}(x^{t,c}, \tilde{t})$ defined in (5);
- $A^{t,\tilde{t}} \in \mathbb{R}^{m \times m}$ is a square matrix with dimensions equal to the number of genes, and each element $A_{g_1,g_2}^{t,\tilde{t}}$ indicates the power of predicting the velocity of gene $g_2$ at time $t$ from the velocity of gene $g_1$ at time $\tilde{t}$;
- the regularization is a combination of the $l_1$ and $l_2$ norms, following the definition of elastic nets [29], with two hyperparameters that control, respectively, the overall amplitude and the ratio between the $l_1$ and $l_2$ norms.

Note that regression requires paired information of the velocities at times $t$ and $\tilde{t}$: since this is not available as each cell is only observed once, we instead use the predicted velocities defined in (5).

After solving the regression problems above, we obtain a family of matrices $A^{t_k,t_{k+1}}$ for any $(t_k, t_{k+1})$. To consolidate these matrices into a global network, we take the sum over all time points:

$$A^{\text{global}}_{g_1,g_2} = \sum_{k=1}^{N-1} A^{t_k,t_{k+1}}_{g_1,g_2}.$$

(8)

**Remark 1 (vector autoregression).**    The linear regression above can be viewed as a regularized first-order vector autoregression (VAR) problem [30], which is a standard approach for forecasting tasks that arise in signal processing. We note that it would be possible to regress $v(x^{t_{k+1}})$ given information for longer lags, such as $\mathcal{T}^{t_{k+1},t_{k-1}}(v(x^{t_{k+1}}, t_{k+1}))$, where the projection is given by the composition of two operators of lag one: $\mathcal{T}^{t_{k+1},t_{k-1}}(v)$ $= \mathcal{T}^{t_k,t_{k-1}}(\mathcal{T}^{t_{k+1},t_k}(v))$. However, as pointed out in [21], the long-term composition of couplings may not be as accurate, and the information through more lags might cause overparameterization of the regression model. Therefore we use the regression problem of the first order only.

**Remark 2 (related approaches with regression).**    There have been alternative approaches which pose linear regression problems in similar fashion, where the predictor and outcome variables can vary. In [7], the Kolmogorov–Smirnov statistic was employed to quantify the global variation of gene expression levels across any pair of consecutive time points, and edges were identified by predicting the Kolmogorov–Smirnov statistic between time $t_k$ and $t_{k+1}$ from the statistic between $t_{k-1}$ and $t_k$. This approach has proven successful for a wide range of tasks, but it loses the granularity by aggregating all time points into one regression problem and fails to characterize the temporal evolution of the network. Another method [13] took a similar approach by solving a linear regression problem per time point, predicting the forward velocity directly from the gene expression levels. While this parameterization does not require the additional barycentric projection as the gene expression levels and velocities are naturally paired within each individual cell, it relies on the assumption of a first order linear system and may fail to detect the connection otherwise. For instance, it may falsely attribute the activation/inhibition of other genes to the housekeeping genes that maintain a high expression level across all time points.

**Remark 3 (supervision via domain knowledge).**    In practice, additional domain knowledge is often available in the form of specific stimuli, transcription factors (TFs), or target genes that are relevant in the specific context considered. This knowledge can be incorporated easily into our framework: for instance, we could use count matrices for only the stimuli and TFs to predict the target genes of interest.

## Temporal gene-velocity relations predict gene regulatory networks

We now describe how we can use time-lagged correlation or, alternatively, Granger causality to infer the underlying GRNs. OTVelo-Corr uses the time-lagged correlation matrix $C$ defined in (6), while OTVelo-Granger relies on the aggregated regression matrix $A^{\text{global}}$ defined in (7) and (8). Each of these two methods outputs a weight matrix that quantifies the strength of regulation between each given pair of genes. To convert the weight matrices into a directed graph, we may apply a threshold to the absolute value of each entry in the weight matrices and set all values below the threshold to zero: this produces a ranked list of edges with nonzero weights.

We emphasize that our algorithm can also produce a dynamic temporal graph model that reveals information within any specified interval: For each time interval $[t_k, t_{k+\Delta k})$ of interest, we can consider the weight matrices obtained by summing only over the times $t_j$ in the interval $[t_k, t_{k+\Delta k})$ in (6) and (8) instead of the full time course from $t_1$ to $t_N$. Thus, the resulting weight matrices measure the strength of the regulations for any pair of genes within the given time interval. This approach can be used to infer temporally changing GRNs in situations where the mechanisms underlying gene regulation vary over time.

OTVelo-Corr and OTVelo-Granger have different advantages and limitations. OTVelo-Corr yields graphs that are usually not sparse, so it is more suitable for cases where the ground-truth network is expected to be dense. OTVelo-Granger, on the other hand, usually returns sparse graphs, especially when the parameter $r$ in (7) is close to one. In terms of computational efficiency, OTVelo-Corr scales much better than OTVelo-Granger with the number of genes. We will revisit and validate these comments further in the results section.

## Preprocessing, normalization, and distance metrics

In this section, we discuss additional details of the proposed pipeline.

**Preprocessing:**  We preprocess raw count matrices by applying the conventional log transformation $\log(x+1)$ to individual counts [31]. Since the preprocessed data will explicitly enter into the distance matrices in the optimal-transport problem (1), it is generally best to apply preprocessing steps that preserve the inherent structure of the underlying biological mechanisms.

**Implementation of optimal transport:**  Next, we describe the implementation details of the entropic regularized fused Gromov–Wasserstein problem 2. The cost matrix $D^{t,\tilde{t}}$ is set to the Euclidean distance between two cells so that $D^{t,\tilde{t}}_{c,\tilde{c}} = \|x^{t,c} - x^{\tilde{t},\tilde{c}}\|$. For the Gromov–Wasserstein term, we follow the setup of SCOT [25] by using the geodesic distance of $k$-nearest neighbor graphs with the number of neighbors determined by $\min\{50, 0.2n(t), 0.2n(\tilde{t})\}$. Each of the three cost matrices $D^{t,\tilde{t}}$, $S^t$, and $S^{\tilde{t}}$ is normalized so that the maximum of its entries equals one. We note that it is possible to use different distance functions in cell space: for instance, empirically, branching structures are often best represented by other distance functions such as the cosine correlation or by using only a small number of marker genes or of highly variable genes. For the two parameters $(\alpha, \epsilon)$ in (2), we note that $\alpha = 0.5$ and $\epsilon = 0.01$ give stable results across all data sets. We always set the marginals $p$ and $q$ to uniform distributions with equal probability for each cell.

**Cell velocities of stimuli:**  In applications where a stimulus is involved, we follow the approach of [12] by creating an artificial gene with velocity equal to one for the time point at which the stimulus occurred and zero elsewhere.

**Linear regression:**  Before solving (7), for each gene, we normalize the expression of all cells across all time points to have standard deviation equal to one. We do not shift velocities to have zero mean as their signs are meaningful biologically and will be used to determine whether a regulation is activation or inhibition.

## Results

### Overview of algorithms, metrics, and hyperparameters

**Algorithms for comparison:**  We compare our method with the following state-of-the-art approaches that are all capable of identifying directed gene-to-gene interactions and quantifying the strengths via a weight matrix:

- GENIE3 [3] infers the regulatory network for each gene using tree-based ensemble methods to predict the expression profile of each target gene from all other genes. The prediction is not signed. The method was originally developed for bulk data and was the top performer in the DREAM4 challenge.
- GRNBoost2 [32] is a more efficient alternative for GENIE3, with acceleration achieved by stochastic boosting algorithms and early stopping criteria. Similar to GENIE3, it utilizes a regression model to infer regulators for each gene and produces a directed and unsigned network.
- SINCERITIES [7] takes time-stamped data and characterizes temporal change in each gene's expression between consecutive time points via the Kolmogorov–Smirnov statistics. The regulatory network is inferred via a linear regression problem with signs determined through partial correlation analysis.
- HARISSA and CARDAMOM [11,12] model the temporal evolution of gene expression levels through a mechanistic model. HARISSA infers the network based on likelihood maximization, and CARDAMOM is a simplified and scalable alternative that exploits the notions of landscape and metastability.

We did not include algorithms that cannot provide directed predictions such as PIDC [6], PPCOR (partial correlation) [33], and the Pearson correlation. In addition, we only considered algorithms that allow for time-stamped data and do not require all cells ordered according to pseudotime such as LEAP [5] and SCRIBE [8].

In Table 2, we list for each of the algorithms we use for our comparison whether it (1) takes time-stamped data to exploit the temporal information, (2) allows for dynamical decomposition of GRNs during different time windows, (3) predicts the sign (type) of regulation, and (4) assumes any particular form of underlying model. Compared to other methods, OTVelo models the temporal evolution of GRNs based on time-stamped data without any explicit assumption of the underlying dynamical model.

**Performance metrics:**  We evaluate the performance of these algorithms on three types of datasets: (1) synthetic networks simulated by HARISSA [11], (2) curated Boolean models simulated in [16], and (3) experimental single-cell RNA-seq datasets from human, mouse,

**Table 2. A summary of the features of the algorithms we use for comparison with OTVelo.**

|  | Time-stamped input | Dynamical inference | Predict signs | Model-free |
|---|---|---|---|---|
| GENIE3 [3] GRNBoost2 [32] | ✗ | ✗ | ✗ | ✓ |
| SINCERITIES [7] | ✓ | ✗ | ✓ | ✓ |
| HARISSA, CARDAMOM [11,12] | ✓ | ✓ | ✓ | ✗ |
| OTVelo | ✓ | ✓ | ✓ | ✓ |

and *drosophila* cell populations. To quantify the performance of each algorithm, we use metrics that are consistent with the original publication associated with each dataset as introduced below:

- AUPRC, AUROC based on unsigned prediction: We compute *Areas Under the Precision-Recall* (PR) and *Receiver Operating Characteristic* (ROC) curves using the true interactions as ground truth and edges ranked from the weight matrices as predictions, ignoring any self-loops.
- AUPRC and AUROC based on signed prediction: We use the metric in [7] to assess the ability to predict the type of regulations correctly. If the sign of an interaction differs from the ground truth, we set the weight of interaction to zero before evaluating the AUPRC values as above.
- AUPRC ratio defined as the AUPRC value divided by the random baseline: If the algorithm yields performance above a random classifier, this ratio should be greater than one.
- Early precision defined as the fraction of true positives among the top $k$ edges, with $k$ determined by the minimum of the number of true positives in the ground truth and the prediction.

AUROC, AUPRC, and Early Precision emphasize different performance aspects. As suggested in [16], AUPRC scores may be more appropriate for sparse networks that have low edge densities, and for this reason we focus more on AUPRC scores. The AUPRC ratio is equivalent to AUPRC but more informative for comparison when AUPRC values are low. Finally, as edges with top weights are of greater importance to practitioners, we follow [16] and use also early precision (which ignores recall) for benchmarking on experimental datasets.

**Selection of hyperparameters:**  For each of the benchmarking algorithms above, we use the default parameters as in their original implementation on their datasets. Our algorithm OTVelo has four hyper parameters: (1) the trade-off parameter $\alpha$ in (2), (2) the entropic regularization coefficient $\epsilon$ in (2), (3) the magnitude $\lambda$ of the regularization term in the regression step in (7), and (4) the ratio $r$ of $l_1$ versus $l_2$ regularization used in (7). The first two coefficients affect the alignment in cell space (and therefore also all subsequent steps), while $(\lambda, r)$ affect only the result of OTVelo-Granger. We demonstrate in S6 Fig how $(\alpha, \epsilon)$ affect the velocity field, while the effect of the regression parameters in OTVelo-Granger on the same dataset is illustrated in S7 Fig.

We use the default hyperparameters $\alpha = 0.5$ and $\epsilon = 0.01$ in OTVelo and choose the two additional hyperparameters $\lambda = 1$ and $r = 0.5$ for OTVelo-Granger. These default values were determined through an extensive hyperparameter search over the synthetic and curated networks. Details on how performance varies with these hyperparameters can be found in the Supplementary S3 Fig and S4 Fig. We note that it is possible to use $k$-fold cross validation to determine optimal values for $(\lambda, r)$ in the regression step and report on the resulting performance metrics for this approach in S5 Fig. Since we did not observe significant performance improvement and since this approach is computationally intensive, we did not implement this option in the package.

## Results for simulated data

We first demonstrate the accuracy of OTVelo by applying it to simulated datasets with a known ground-truth network. Such datasets are often created by simulating stochastic differential equations that incorporate the structure of the given network and therefore admit an inherent ground truth that we can use when evaluating the performance of a given algorithm. Here we test our algorithm on datasets from two different simulators: (1) synthetic networks generated by the mechanistic model-based simulator HARISSA [11,12] and (2) curated networks from known biological processes simulated by BoolODE [16]. For each simulator, we use datasets that are already available and have been employed for benchmarking GRN inference algorithms. The networks and the performance of the GRN inference algorithms are summarized in the subsequent subsections.

**OTVelo-Granger outperforms on most synthetic network datasets.** We first apply our algorithm to nine synthetic networks that were generated from the mechanistic model described in [11] and are illustrated in the first row of Fig 3:

- FN4: Four genes with a branching structure and an inhibition feedback loop;
- CN5: Five genes with a cycling structure;
- FN8: Eight genes with branching and feedback loops;

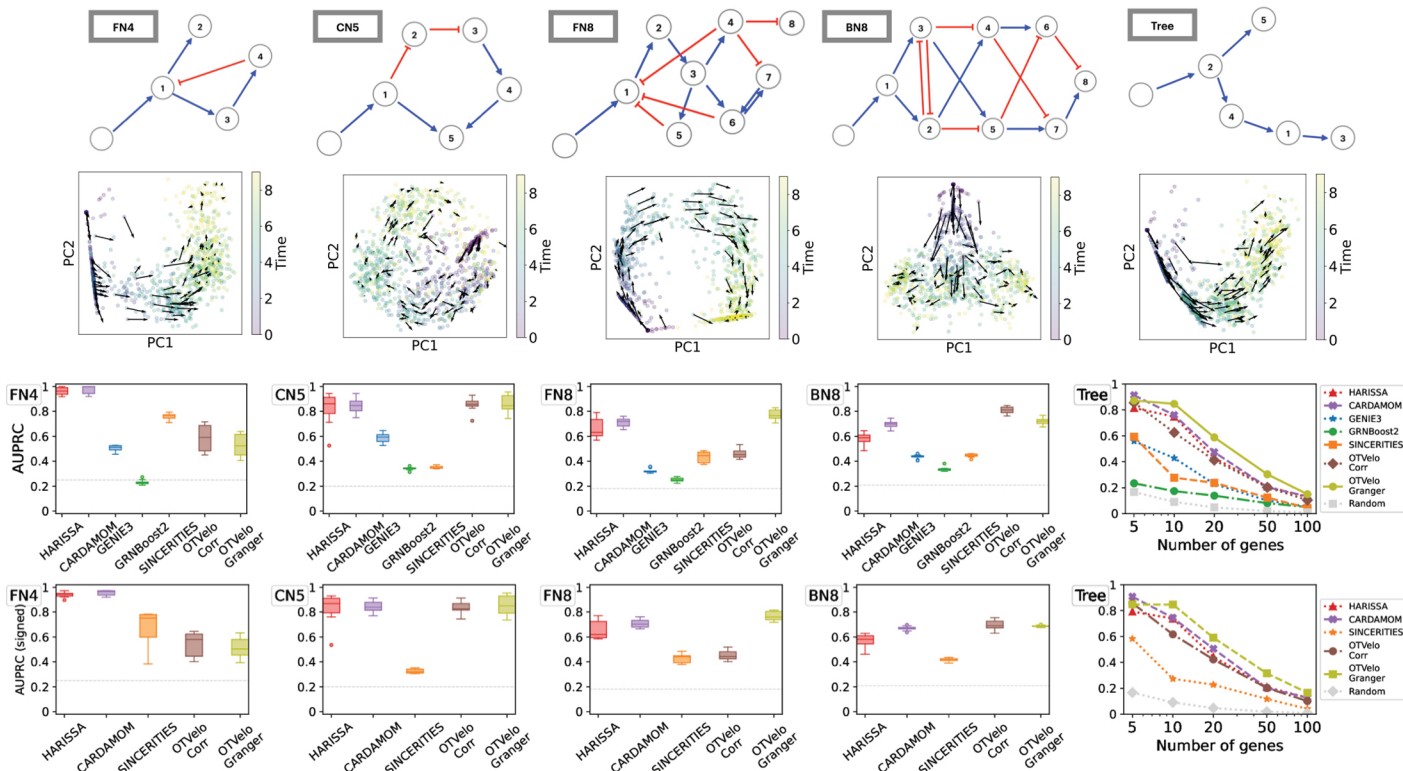

**Fig 3. Results on simulated data from Harissa [11].** First row: Networks used for subsequent tests, where each network starts with a stimulus, with blue edges indicating activation and red edges indicating inhibition, reproduced according to Ventre et al. [12] after excluding self-loops. Second row: first two principal components of cells and velocity field inferred by our methodology. The third and fourth rows show performance of 6 algorithms, measured by AUPRC and AUPRC on signed prediction. Note that GENIE3 and GRNBoost2 cannot predict the sign of regulation and are removed from the fourth row. The gray dashed line indicates the baseline that correspond to a random classifier.

- BN8: Eight genes with branching trajectories in the resulting cell trajectories;
- Trees: Five tree-like networks with 5, 10, 20, 50, and 100 genes, respectively.

Each of these networks contains a stimulus that is switched on at the first time point and should therefore serve only as a source of regulation at the beginning. To account for this, we define an artificial gene that has velocity one at the first time point and zero otherwise, and use it as the predictor in the linear regression in (7) as outlined in Remark 3 on supervision via domain knowledge.

For each network, we use 10 independent datasets generated by Ventre et al [12]. Each dataset contains 10 time points, and 200 cells were drawn at each time point. We first show that the gene velocity captures the temporal dynamics by visualizing the velocity field. In the second row of Fig 3, for one dataset simulated from each network, we compute the gene velocity as defined in equation (4) after replacing the full gene expression $x$ by its first two principal components. The gene velocity is then visualized via the vector field defined at single-cell resolution.

We summarize the results through the area under the precision-recall curve (AUPRC) in Fig 3. As reported by Ventre et al. [12], GENIE3 and SINCERITIES do not perform well on these tasks. We observed that our correlation approach ranked the best on datasets BN8, while OTVelo-Granger resulted in best performance on FN8 and Tree structure with number of genes greater than 5. In addition, we quantify the ability to infer the signs (activation or inhibition) correctly by the signed AUPRC, as visualized in the fourth row of Fig 3. We observed consistent agreement with the unsigned AUPRC, indicating our algorithm's capability to capture the correct type of regulation. In particular, OTVelo-Granger outperforms on signed AUPRC on all datasets except the two smallest datasets (FN4 and tree with only 5 genes). The fact that our method performs less well on simple network is likely due to the slow temporal variation in gene expression levels, while OTVelo is designed for capturing regulation when gene expression changes rapidly between adjacent time samples through time-lagged correlation. In contrast, our method achieves better performance for those complex CARDAMOM datasets, while model-based methods such as HARISSA and CARDAMOM can get stuck locally and fail to converge to the true model.

**Effect of simulation parameters.** Next, we investigate whether performance is affected when (1) the number of cells at each time point, (2) the length of time gap between each pair of consecutive time points, or (3) the length of the measurement period are changed. Following the setup of CARDAMOM [12], we simulate 10-gene tree structure networks with one parameter varying at a time and report the AUPRC values in Fig 4. Similar to the observations made in [12], we find that the number of cells per time impacts the performance less than the other two factors, while the length and lag of time points significantly affects the accuracy. Our methods perform best at a critical time lag of approximately 10, which suggests that the time scale of lagged correlations plays a critical role here, possibly due to the fact that our method relies on correlation (or causality) between consecutive time points.

**OTVelo-Corr outperforms on most curated networks.** As pointed out in [16], while the synthetic network models presented above are useful for generating simulated data and for benchmarking, these networks do not necessarily reflect the characteristics of real single-cell datasets. Therefore we tested our methods on published Boolean models which were built to reflect real cellular processes. We use four published models from [16] that correspond to Mammalian Cortical Area Development (mCAD), Ventral Spinal Cord Development (VSC),

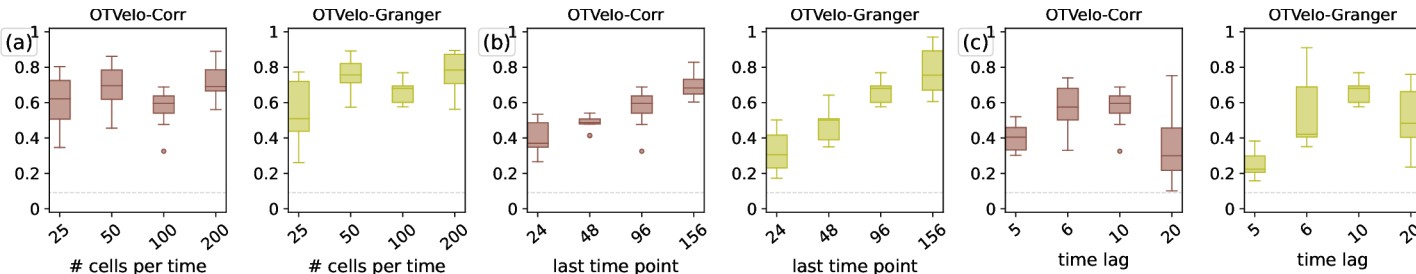

**Fig 4. AUPRC for predicting directed edges for the two OTVelo methods, profiled on 10-gene tree-structure networks with different data collection parameters.** (a) Performance as a function of the number of cells per time point, while keeping the same time points. (b) Performance as a function of the length of the measurement period, while keeping the same gap between time points and the same total number of cells. (c) Performance as a function of the gap between time points, while keeping the same final time point and the same total number of cells.

Hematopoietic Stem Cell Differentiation (HSC), and Gonadal Sex Determination (GSD), respectively.

For each model, we use the ten datasets simulated and provided by [16], each contains 2,000 cells and multiple branches. As the datasets are not time-stamped, we first assign the time points following the approach of [7,16] by binning the pseudotimes provided along with the datasets. The total number of time points is given by the values that gave best performance for SINCERITIES as reported in the supplementary material of [16]: we use 10 time points for mCAD, 5 for VSC, 20 for HSC, and 6 for GSD. We then run the algorithm separately on each branch and combine the resulting inferred GRNs into a single aggregated graph as further described below.

The results are shown in Fig 5. Since the AUPRC values are typically low, we follow the approach in [16] and report instead the AUPRC ratio, which is equal to the AUPRC divided by the random baseline: this quantity should be greater than one if the performance is better than a random classifier. For OTVelo, we combine different branches by adding the branch-specific weight matrices across all branches and then calculating the resulting graph by thresholding. For HARISSA, CARDAMOM, and SINCERITIES, we follow the approach in BEELINE [16] and combine the branch-specific graphs by taking the maximum of the absolute values across all branches. OTVelo-Corr performs consistently very well for both signed and unsigned inference across all datasets, while OTVelo-Granger does not perform as well, especially on VSC and HSC which are the two datasets with more than two branches.

**Effect of dropouts.** To assess the effect of dropouts on performance, we run the algorithms also on the datasets with 50% and 70% dropout rates that were provided in [16] without recomputing the pseudotime. As shown in the results in Fig 6, dropouts hurt the performance of almost all algorithms, including ours, unless the performance was already below the random baseline. In these scenarios, our algorithm consistently outperforms the other methods on mCAD, HSC, and GSD regardless of dropout rate.

**Effect of splitting branches.** Optimal transport-based methods have the capability to follow cells as they transition along different paths. We therefore also test whether it is necessary to compute pseudotimes and split the dataset into different branches before running the GRN inference algorithm by comparing the following two approaches:

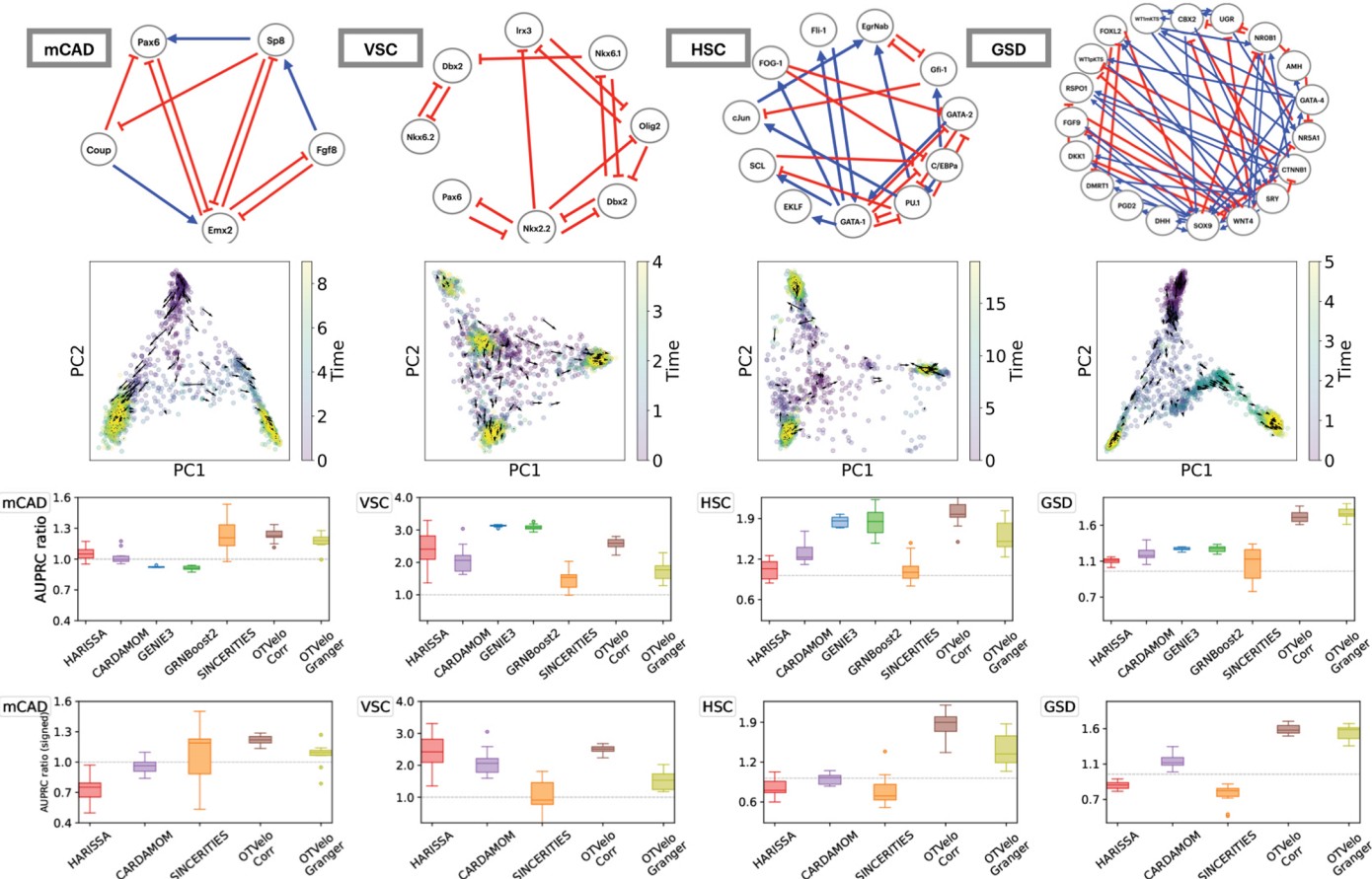

**Fig 5. Results on curated networks based on simulation from BoolODE as used in BEELINE [16].** The top row shows the four curated networks that correspond to actual biological processes (red edges indicate inhibition, and blue edges indicate activation), reproduced from BEELINE [16] after removing self-loops. The second row shows the velocity field after projection onto the first two PCA modes, and the third and fourth row contain the AUPRC ratios for, respectively, unsigned and signed inference; note that GENIE3 and GRNBoost2 are removed from the signed AUPRC panel due to their inability to predict the type of regulation. The dashed gray line is the baseline that corresponds to a random classifier.

Split (each branch separately): Once branches have been identified through Slingshot, we run our algorithm separately on each branch, create a graph for each of them, and finally combine these graphs into a single aggregated graph by taking the maximum absolute value of edge weights over all branch-specific graphs and assigning a sign according to the maximum value. This approach was used for the results shown in Figs 5 and 6.

Combined (all branches together): We use the entire dataset with time stamps given by the actual simulation times (instead of pseudotime) and bin time points by quantiles as before. Applying our algorithm directly yields an inferred GRN for the full dataset.

The results can be found in Fig 7, where performance is again quantified via unsigned and signed AUPRC ratios. OTVelo-Corr is less accurate when using the combined data. In contrast, OTVelo-Granger seems to be robust regardless of which approach is used. We note that the "Split" approach requires solving the full optimal-transport plan once for each branch, while the "Combined" approach requires only a single solve.

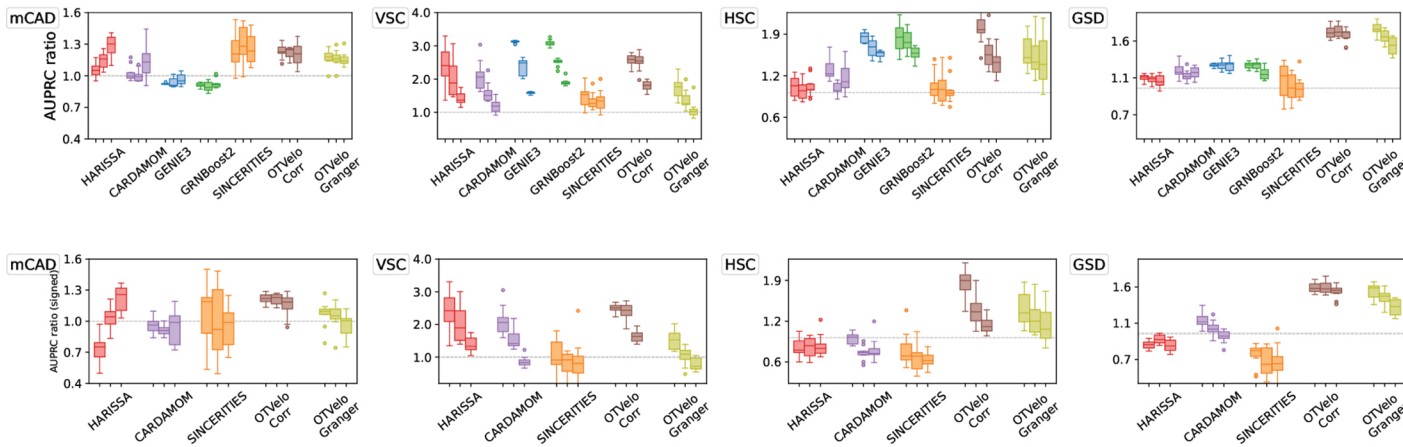

**Fig 6. Results on curated network models with different levels of dropouts, quantified by AUPRC on unsigned (top row) and signed (bottom row) predictions.** Each method is represented by three box plots that correspond (from left to right) to 0%, 50%, and 70% dropout rates.

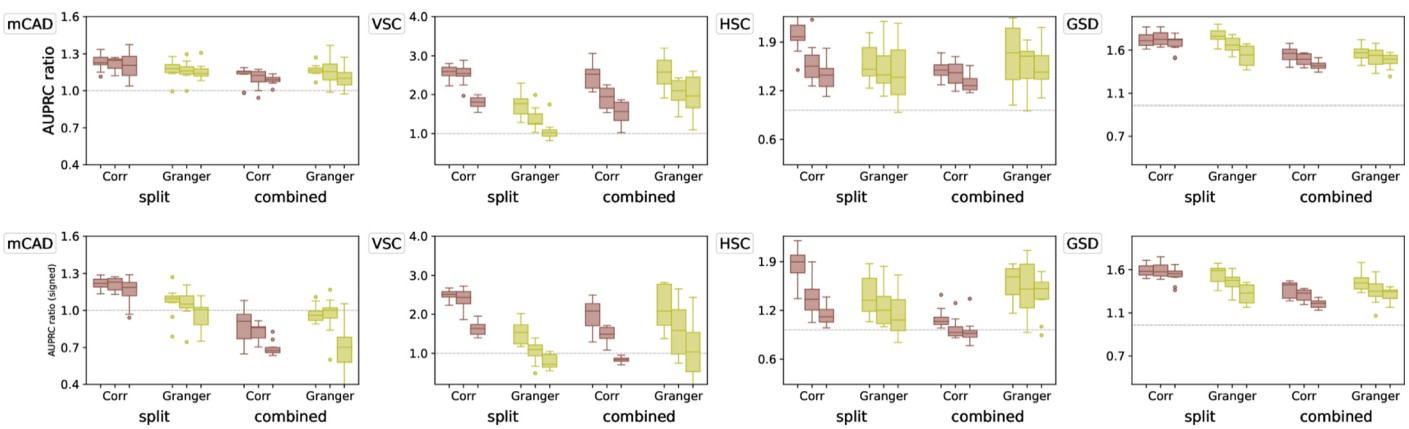

**Fig 7. Performance of our algorithm for the "Split" and "Combined" setups described in the main text as quantified via unsigned (top row) and signed (bottom row) AUPRC ratios on curated datasets with 0%, 50%, and 70% dropout rates (from left to right boxplot).**

**Effect of disproportionate cell-type representation.** Next, we discuss the situation where the fractions of cells in each cell type (or branch) vary over time. Since the percentage of each branch in the datasets we used does not vary significantly across time points, we first create imbalanced samples by downsampling the first branch at later times and the second branch at earlier times. Afterwards, we run OTVelo on the resulting dataset using either (1) uniform weights for all cells or (2) prescribing a reweighed marginal distribution that compensates for the varying fractions of cells in each branch to guarantee that the total weight of each branch is independent of the time point. The second approach ensures that each branch can be traced out properly, while the first approach with uniform weights across all cells may erroneously map cells to overrepresented branches. Our results show that using reweighed marginals helps preserve branching structure as indicated by the velocity field (Fig 8, top right) compared to using uniform weights (Fig 8, top left). This is also reflected in the better performance quantified by the AUPRC ratio (Fig 8, bottom), especially for the two multi-branching datasets VSC

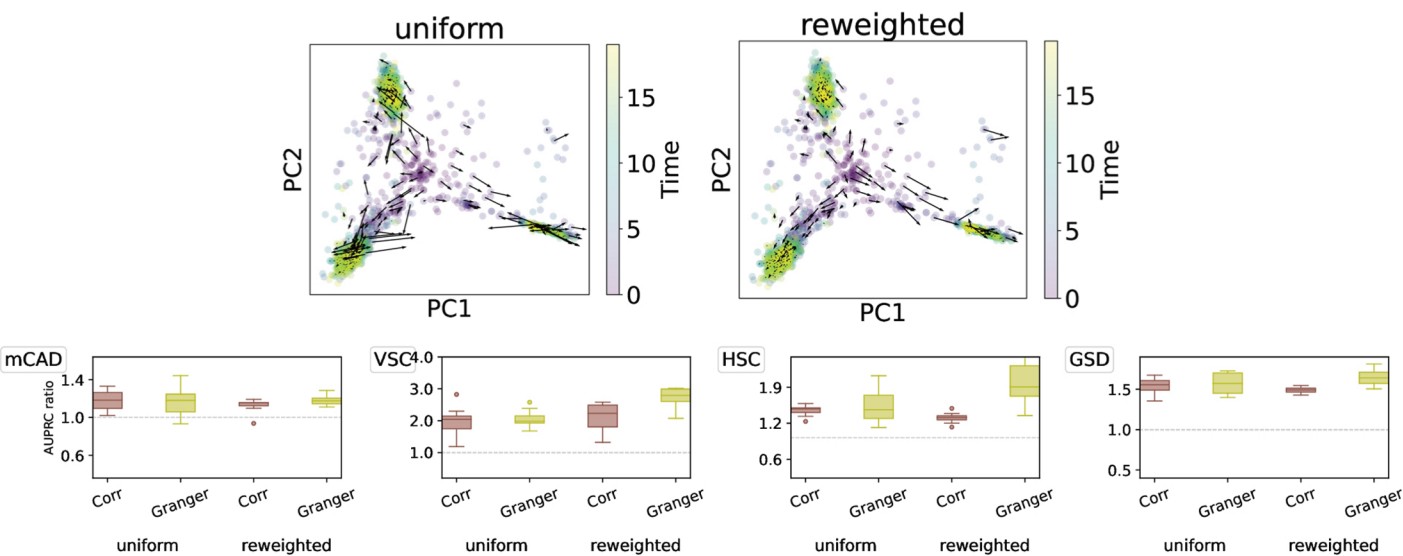

**Fig 8. Performance of OTVelo on unbalanced datasets with uniform and reweighed marginals.** Top: velocity field for HSC with uniform weights (left) and reweighed marginals (right). Bottom: AUPRC ratios for uniform weights and reweighed marginals assigned based on the proportion of cells in each branch.

and HSC. These findings indicate that our algorithm can also be used for unbalanced datasets caused by either sampling variability or cell proliferation. In real datasets, such information is often partially known either through cell-type annotation or gene signatures of proliferation and apoptosis [21]. We recommend incorporating such information into the computation of the coupling matrices whenever it is available and use them for downstream GRN inference through either correlation or Granger causality.

## Results on experimental datasets

In this section we test our algorithm on nine experimental single-cell gene expression datasets and compare the inferred GRNs with the ground truth when applicable. First, we benchmark our algorithm on five standard experimental datasets as used in BEELINE [16]. The scGEM [34] and Kouno [35] datasets contain only one lineage, while the third dataset [36] exhibits branching into two different cell types. For these three datasets, the time stamps at which data were collected is available. The last dataset captures the dynamics of *Drosophila* neuroectodermal tissue [37]: with count matrices for 5000 genes, this dataset is larger, and information about the time stamps is not directly available and instead learned from a neural network as discussed in [37].

**OTVelo outperforms on three out of five standard benchmarks.** First, we examine the performance of OTVelo on five standard benchmarks from BEELINE [16] and STREAM-LINE [38], namely, mouse hematopietic stem cells (mHSC [39]), mouse embryonic stem cells (mESC [40]), mouse dendritic cells (mDC [41]), human mature hepatocytes (hHEP [42]), and human embryonic stem cells (hESC [43]). Since edges with high weights are of greater importance to practitioners, we quantify performance by early precision ratio (fraction of true positives in top edges, normalized by early precision of a random predictor).

We use pseudotime computed via Slingshot [2] to assign time points. Genes that are either significantly varying TFs (with *p*-value less than 1% after Bonferroni correction) or are among the top 500 (or 1000) highly variable genes are included. Ground truth networks come from

either cell-type specific ChIP-seq or the STRING database (which may include indirect causal relationships in addition to direct regulation). For methods that require binning cells according to time (HARISSA, CARDAMOM, SINCERITIES, OTVelo), we perform a grid search over 4 to 15 bins divided according to quantiles for each of them and record only the value of the metric that corresponds to the best performance. As shown in Fig 9, the performance metrics on datasets with cell-type specific ChIP-seq ground truth are typically close to 1 (and therefore perform similarly to a random predictor) with OTVelo-Granger achieving best performance on mHSC, mDC, and mESC for top 500 variable genes combined with TFs. With top 1000 variable genes, OTVelo-Granger outperforms on mHSC, mESC, hHep, and hESC.

**OTVelo reveals consistent temporal dynamics on scGEM reprogramming data.** Next, we consider a single-cell dataset from human cellular reprogramming, where cells were collected every 8 hours and the expression of 34 key cell-stage specific markers were measured. Fig 10 shows the resulting network divided according to time points, where blue edges indicate activation, and red edges indicate inhibition. Although no ground truth network is available, the edges suggest consistent information regarding repression of somatic genes (blue group) and activation of pluripotent genes (yellow) as reported in the original publication [34]. Moreover, as the velocity only captures the rate of change and ignores genes that do not vary much over time, most of the genes that were not labeled in the original publication (purple group in Fig 10) were not assigned any edges. Fig S8 Fig shows that HARISSA, CARDAMOM, and GENIE3 produce interaction graphs consistent with ours, while SINCERITIES does not consistently assigns repression and activation to the same gene groups.

Another benefit of our method is the decomposition of the inferred GRNs into GRNs for each time interval by using the individual matrices $A^{t_k, t_{k+1}}$ from (7) in OTVelo-Granger (or, alternatively, each of the terms inside the summation over $k$ in (6) in OTVelo-Corr). The resulting dynamic GRN shows a consistent path of information that starts with the repression of somatic genes, continues with activation of pluripotency genes, and ends with no interactions occurring during the last stage IPS to ES. We note that other methods do not provide a dynamic decomposition of the inferred GRNs.

In addition to the qualitative observations mentioned above, we also benchmark our method based on two types of ground truth: (1) networks validated by experiments are

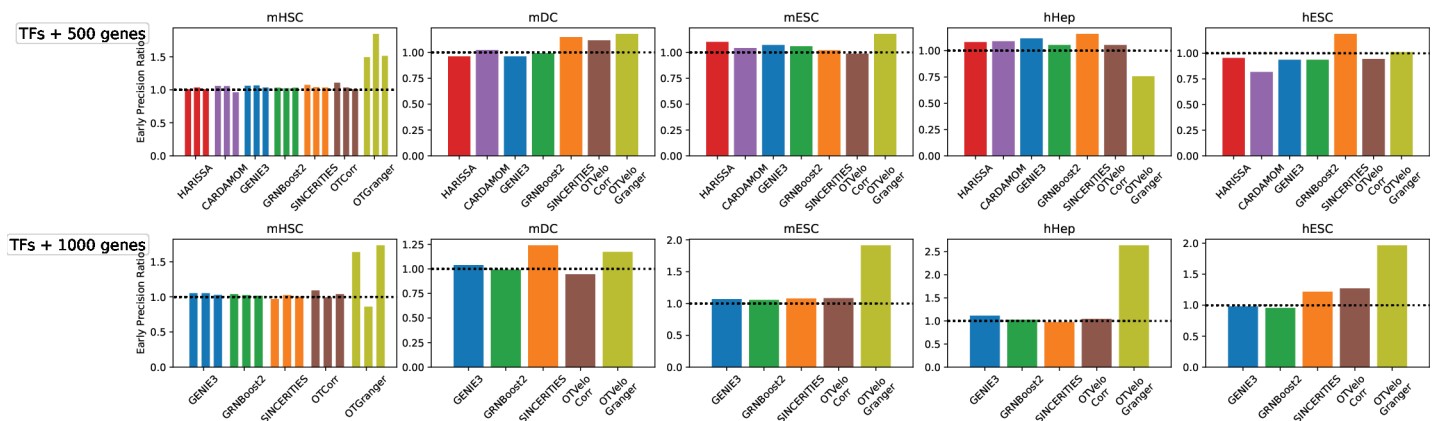

**Fig 9. Early precision ratio of five real datasets compared against ground truth based on cell-type specific ChIP-seq, using either 500 or 1000 top variable genes, combined with significantly varying TFs.** Since mHSC includes three branches, we perform the task on each branch separately as in BEELINE and plot them on the same graph in the first column. HARISSA and CARDAMOM were not included in the second row due to prohibitive computational time.

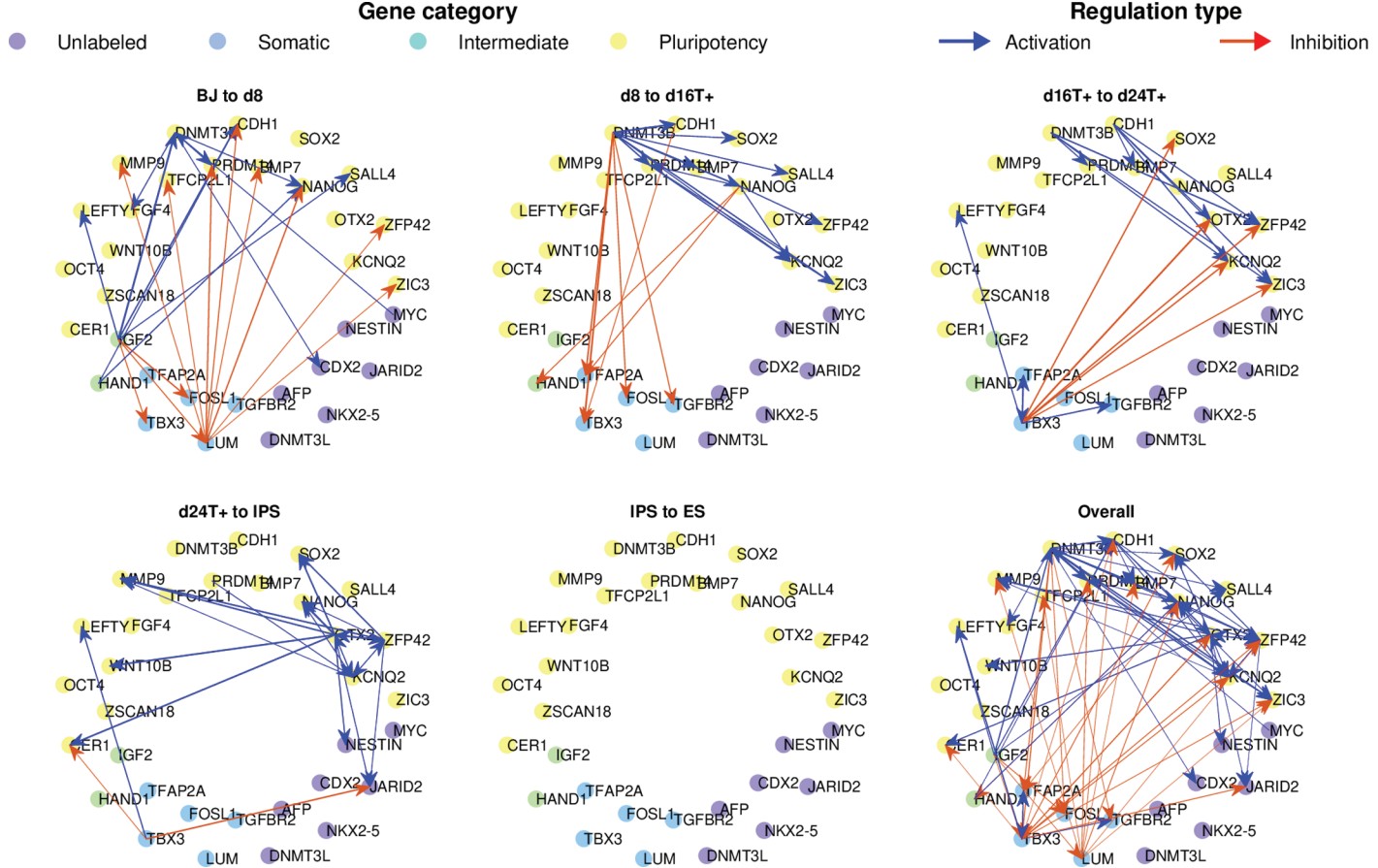

**Fig 10. Breakdown of the GRNs inferred by OTVelo-Granger from the scRNA-seq scGEM dataset [34], where only edges with weights in the top** 1% **in each time interval are visualized.**

obtained from a database specifically for human fibroblasts undergoing reprogramming as in Hamaneh et al. [44], and (2) STRING [45] networks which may include functional relations. The AUPRC and AUROC metrics provided in Table 3 show that our correlation approach is either the best (for the STRING network) or second best (for the experimental network, second to GRNBoost2).

**OTVelo achieves best AUPRC for signed GRNs on THP-1 differentiation data.** Next, we apply our algorithm to infer GRNs from the dataset [35] collected during THP-1 human myeloid leukemia cell differentiation (this dataset was also used in [7]). The time-stamped data [35] were sampled at 8 time points (and specifically after 0, 1, 6, 12, 24, 48, 72, and 96 hrs), and each time point has 120 cells with 45 TFs profiled. This dataset provides a good benchmark because a signed subnetwork of 20 TFs has previously been constructed from experiments [46] and can be used as 'ground truth' when evaluating the AUPRC and AUROC metrics. We run the algorithm on all 45 TFs and evaluate the metrics based on the regulatory edges amongst the 20 TFs used in [46]. The metrics for both signed and unsigned predictions are shown in Table 4. Note that we provided only unsigned AUPRC and AUROC for GENIE3, since it cannot predict whether edges correspond to activation or inhibition.

**Table 3. Results on scGEM with ground truth determined by (1) cell-type specific networks validated by experimental data [44] and (2) STRING [45]. The random baseline for AUPRC is 0.020 for the experimental network and 0.058 for the STRING network.**

|  | experimental network | | STRING network | |
|---|---|---|---|---|
|  | **AUPRC** | **AUROC** | **AUPRC** | **AUROC** |
| GENIE3 | 0.027 | 0.592 | 0.125 | <u>0.691</u> |
| GRNBoost2 | **0.033** | <u>0.616</u> | 0.147 | **0.709** |
| SINCERITIES | 0.018 | 0.474 | 0.058 | 0.511 |
| HARISSA | 0.028 | 0.586 | 0.078 | 0.619 |
| CARDAMOM | 0.024 | 0.566 | 0.081 | 0.614 |
| OTVelo-Corr | <u>0.031</u> | **0.667** | **0.167** | 0.666 |
| OTVelo-Granger | 0.020 | 0.480 | <u>0.150</u> | 0.637 |

**Table 4. Results on the THP-1 single-cell dataset [35], compared with the ground truth constructed as in [46]. For each metric, the best value is indicated in bold, and the second best is underscored.**

|  | AUPRC | signed AUPRC | AUROC | signed AUROC |
|---|---|---|---|---|
| GENIE3 | 0.26 | - | 0.47 | - |
| SINCERITIES | **0.32** | 0.20 | **0.68** | **0.61** |
| HARISSA | 0.19 | 0.10 | 0.49 | 0.39 |
| CARDAMOM | 0.21 | 0.13 | 0.49 | 0.39 |
| OTVelo-Corr | 0.29 | <u>0.22</u> | <u>0.60</u> | 0.53 |
| OTVelo-Granger | <u>0.30</u> | **0.29** | 0.55 | <u>0.54</u> |

Overall, OTVelo-Corr and OTVelo-Granger achieve good performance on unsigned prediction. When including the sign, our two frameworks achieve the best AUPRC suggesting the effectiveness of our method in identifying the regulation type.

**OTVelo reveals consistent temporal dynamics on mouse data with stimulus**   We further test our approach using the time-stamped dataset [36] obtained by scRNA-seq of a retinoic acid (RA)-induced differentiation of mouse ES cells. This dataset captures the transition from pluripotent embryonic stem cells towards two cellular lineages (namely ectoderm- and extraembryonic endoderm-like cells) as visualized in Fig S9 Fig. Samples were collected at 9 non-uniformly spaced time points (0, 6, 12, 24, 36, 48, 60, 72, and 96 hrs) with a total number of 3456 cells being sequenced.

We preprocess the data exactly as in the original publication [36] by first selecting cells whose total number of UMI counts exceeds 2000, resulting in a set of 2449 cells, where each time point contains between 137 and 335 cells, and normalize counts within each cell by their respective total UMI count number. Finally, we follow the process outlined in [12] and restrict our analysis to a panel of 41 genes that are key marker genes for pluripotency, post-implantation epiblast, neuroectoderm, and extraembryonic endoderm.

To validate our results, we compare a subnetwork to the ground truth reported in [12], which includes unsigned edges pointing out from the RA stimulus and the genes Sox2, Pou5f1, and Jarid2. In Fig 11, we quantify the accuracy of the GRN predictions via the ROC and the precision-recall curve. OTVelo-Granger achieves best AUPRC, while OTVelo-Corr ranks third, tied with HARISSA.

Fig 11 indicates that none of the algorithms performs particularly well on this dataset. Two possible reasons are: (1) The available ground truth curated from [12] consists only of regulations associated with four resources (RA, Sox2, Pou5f1, and Jarid2) for which reliable prior information is available, which does not necessarily reflect biological truth and therefore limits the power of performance quantification. (2) As noted [12], the original experimental

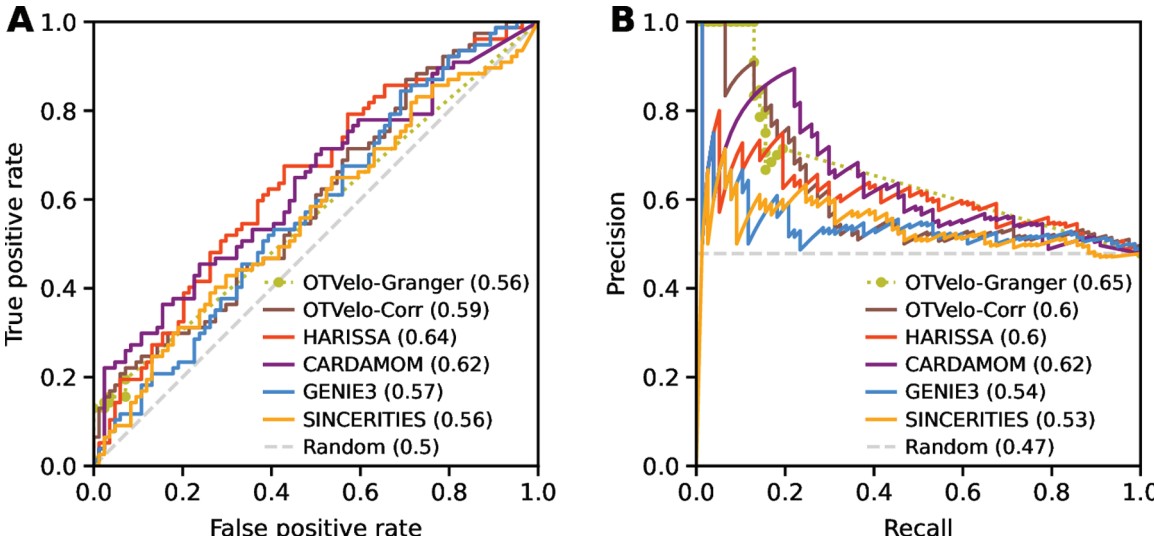

**Fig 11. ROC (panel A) and precision-recall curve (panel B) for the mouse ERS dataset [36].** OTVelo-Granger has the best AUPRC. Both OTVelo methods have accurate prediction for 5-10 edges that are assigned largest weights, indicated by the short plateau in the upper left corner of panel (B).

design is highly unbalanced, since five out of nine time points were collected during the early phase that barely captures the differentiation process (Fig S9 Fig).

To qualitatively investigate the goodness of our results, we visualize the dynamic GRNs predicted from OTVelo-Granger in Fig 12. More edges are identified at the very first and last time points where the reaction happens at a faster rate. No edges were identified during the time intervals from 12hrs to 36hrs, likely because the dynamics is quite slow at the beginning (this is also visible in Fig S9 Fig, which contains the PCA reduction of the dataset over time). We note that these results are consistent with [12,Fig 6] where edges were included at the time point for which they were detected in CARDAMON with the largest intensity.

**OTVelo infers transcription factors and target genes in *drosophila* neuroectoderm data**
To demonstrate the ability of OTVelo to handle large-scale datasets, we apply it to the recent *Drosophila* (fruit fly) dataset published in [37]. Single cells are profiled over 20 hours during embryonic development, and the continuous development age was learned via a deep neural network. We focus on a subsample of cells annotated as neuroectoderm using scRNA data from 6 to 18 hours, during which cells undergo diversification and differentiation, resulting in 8 different cell types. We divide the data into 1hr time windows according to the developmental age, resulting in 12 windows with 1200 cells sampled within each time window. We include the top 5000 variable genes in the computation.

Due to the high dimension, we only run the correlation algorithm, since the linear regression is taking a significant amount of time. As in [7], we define the in-out degree of each gene as:

$$\deg(g) = \frac{\sum_{\tilde{g} \neq g} |C_{\tilde{g}, g}|}{\sum_{\tilde{g} \neq g} |C_{g, \tilde{g}}|}, \tag{9}$$

where $C$ is the correlation matrix as defined in 6. The numerator measures the extent to which a gene might be regulated by other genes, while the denominator reflects how likely

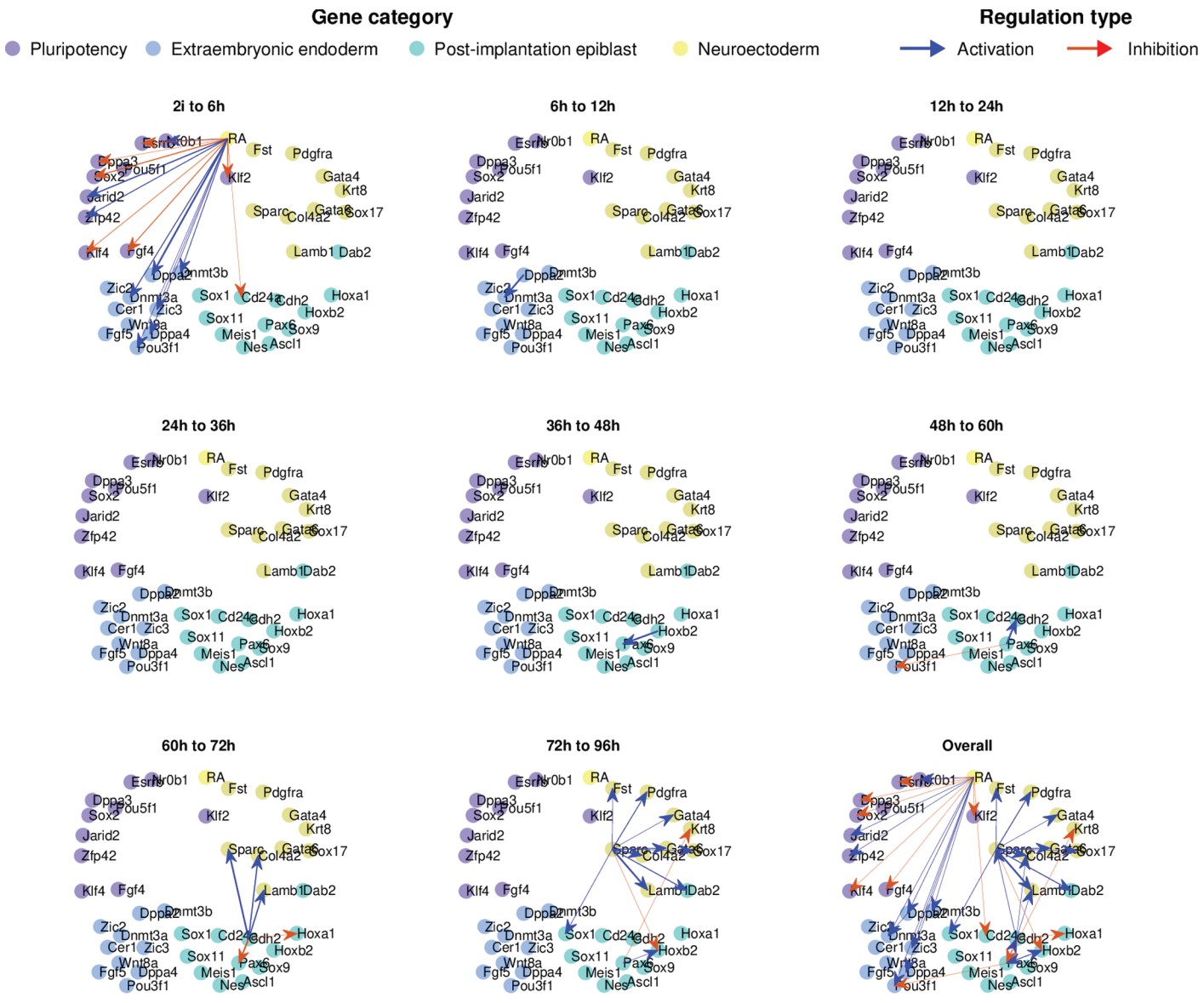

**Fig 12. Dynamic GRNs inferred from OTVelo-Granger for the mouse ERS dataset [36].**

a gene regulates other genes as a transcription factor. Upstream genes will typically result in $\deg(g) < 1$, while downstream genes may have $\deg(g) > 1$.

Fig 13 summarizes our results. We see that $\deg(g)$ is a meaningful metric to quantify the likelihood for a gene to be a transcription factor or a target gene: For instance, a number of transcription factors have the ratio less than one, including *cas*, a known temporal neuroblast transcription factor, and *sv*, a transcription factor that is involved in the development of sensory organs. In contrast, two synaptic genes, *cac* and *Syt1*, receive a ratio greater than one. Fig 14 demonstrates that the in-out degree is robust even when thresholding the $C$ matrix. We propose to use this metric to nominate new potential transcription factors and target genes.

We emphasize that our correlation approach is the only algorithm considered in this work that can be applied efficiently to a dataset of such scale (top 5000 highly variable genes and

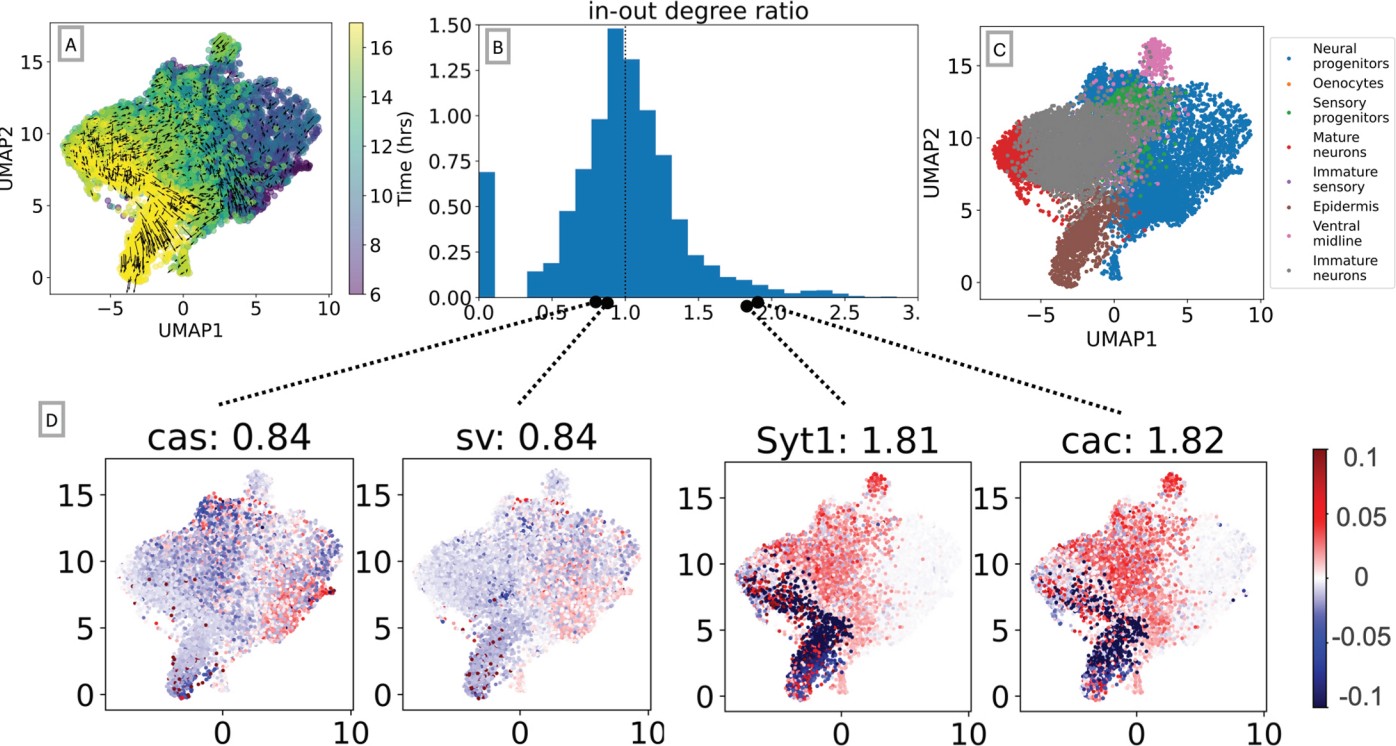

**Fig 13. Results for the *drosophila* neuroectoderm dataset [37].** (A) First two UMAP coordinates of single-cell data along with the field learned via optimal transport. (B) Histogram of the in-out degree ratio as defined in (9). (C) First two UMAP coordinates colored according to annotated cell types. (D) Degree ratio of four genes and their velocity profile over the cell spaces: *cas* and *sv* are enriched in neural progenitors and sensory progenitors that primarily exist in earlier time points, resulting in a ratio of 0.84<1; *cac* and *Syt1* are two markers that are enriched when neurons mature, so they are assigned ratios greater than one, as indicated in their captions.

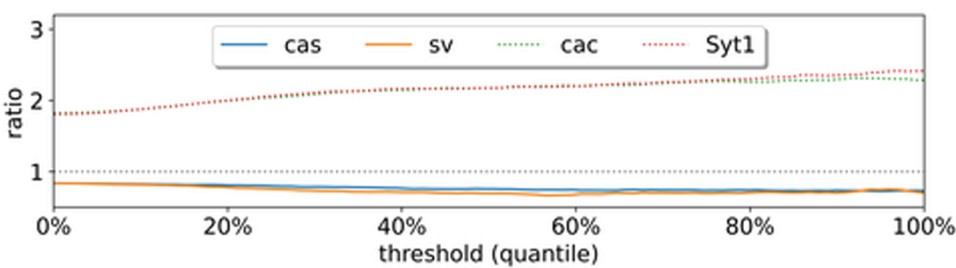

**Fig 14. In-out degree as defined in (9) after thresholding the correlation matrix.**

more than 10,000 cells after subsampling from the cells and selection of variable genes). It was computationally intensive to run other existing algorithms, including GENIE3, HARISSA, and CARDAMOM. Simpler options that are computationally inexpensive such as the Pearson correlation do not provide directed prediction. SINCERITIES without predicting the regulation type can also be executed within a reasonable amount of time, however the prediction of signs requires computing the partial correlation, which is done between any pair of genes and requires prohibitive computational time compared to the regression when the number of genes is large.

To further investigate the computational efficiency, additional experiments were performed on the *drosophila* dataset with 12 time points. We considered two scenarios: (1) We fix the number of cells (1200) per time point, and vary the number of genes within $\{10, 25, 50, 100, 250, 500, 1000, 200\}$. (2) We fix the genes of interest to the top 100 variable genes for each cell at any time point, and vary the number of cells in the subsample from $\{100, 200, 300, 500, 800, 1000, 1200\}$. All experiments were performed on a laptop equipped with an Apple M1 chip (8 GB of unified memory) and 8 cores (4 high-performance cores and 4 efficiency cores), and we compared our method directly to the implementations for the other algorithms supplied in the original publications. The wall clock time (unit: second) are reported in Fig 15. The computational bottleneck of OTVelo are the sequence of fused Gromov–Wasserstein OT solves in cell space, and its computational complexity is known to grow relatively quickly with respect to sample size [24], resulting in fast growth of time as functions of cells as shown in Fig 15 (right), which can be alleviated by utilizing more efficient solvers such as quantized Gromov–Wasserstein [47]. On the other hand, it is insensitive to the number of genes because the associated computational costs can be negligible as confirmed by Fig 15 (left). As a result, when number of genes exceeds 1000, OTVelo-Corr requires the least computational time among all other methods.

## Discussion

OTVelo employs a two-step approach to infer gene-regulatory networks from time-stamped single-cell RNA sequencing data. First, we predict past and future states of individual cells via an optimal-transport plan, which then allows us, via a finite-difference scheme, to calculate gene velocities for each cell at each time point. Second, we infer gene-to-gene interactions across consecutive time point by computing, and thresholding, time-lagged correlation or Granger causality of the gene velocities we computed in the first step. In particular, our approach integrates optimal transport along with velocity estimates to infer GRNs for each time interval. We demonstrated the accuracy and efficacy of the proposed OTVelo framework through its application to simulated, curated, and experimental datasets. We also showed that our algorithm is applicable to situations where only pseudotimes, but not the actual time stamps, are available for the scRNA data.

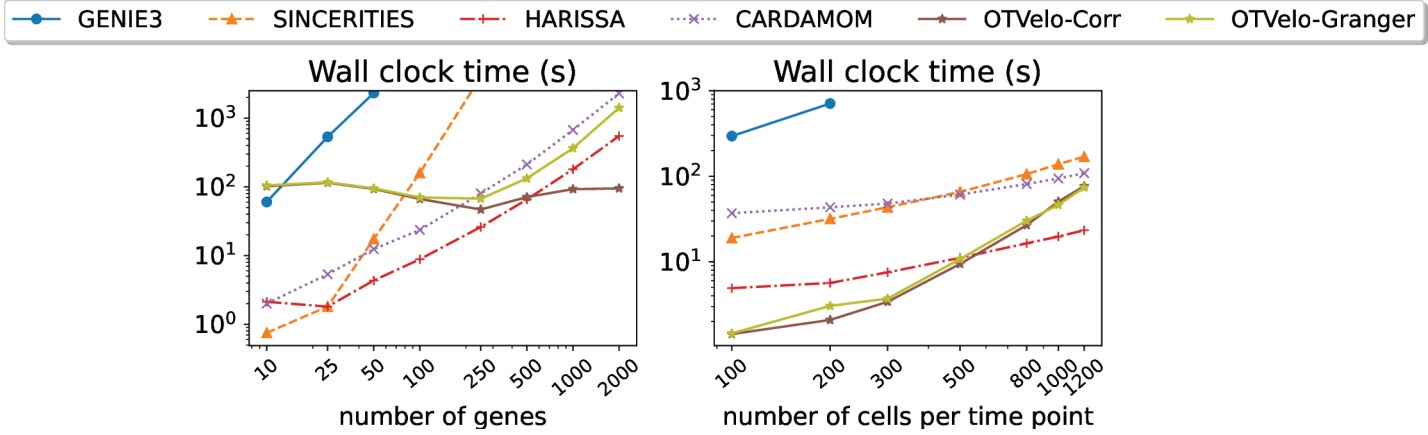

**Fig 15. Computational time for each algorithm as functions of (1) number of genes, with a fixed number of 1200 cells per time point (left); or (2) number of cells per time point, with a fixed number of 100 genes.**

It may be possible to incorporate OTVelo into other existing approaches. For instance, the approaches described in [8,20] rely on RNA velocity measurements, which may not be available if the dataset was not sequenced deep enough to obtain unspliced counts. Our gene velocity estimates could potentially replace RNA velocity in these algorithms.

There are several possible extensions of the OTVelo framework. For instance, our approach can be extended to (1) incorporate different time lags (similar to [5]) instead of restricting to the preceding time point only, (2) use partial correlation instead of correlation to control for other confounding variables, and (3) generalize Granger causality beyond linear regression to account for nonlinear interactions.

It would be interesting to see whether the current OTVelo framework, which relies on scRNA data, could integrate other modalities such as single-cell chromatin accessibility (scATAC-seq). Similar to SCENIC+ [48], this information could be used to determine which gene can be transcribed at a certain time point and could aid in feature selection as the number of genes in real datasets is often too high to be directly used in any algorithm. A second aspect that we did not consider here is the effect of cell-type heterogeneity where the number of cells from each type is not equally balanced due to sample variation and/or cell proliferation or death: similar to [49], using unbalanced optimal transport might help address these challenges.

Another aspect of potential work is to incorporate a rigorous statistical analysis into the framework, as the GRN inference is always performed based on limited samples at a limited number of time points. A first step would be to provide some error quantification, such as 95% confidence intervals, to the edge weights assigned for each regulation relation. Secondly, the framework can be generalized to hypothesis testing to determine whether the GRN depends on time, context, and/or species. Such problems still remains challenging as the samples can be highly correlated either within or across different time points, and suitable assumptions have to be made in order to provide error quantification for networks.

By addressing such challenges and further extending the framework, OTVelo has the potential to become an even more powerful tool, enabling more accurate and comprehensive insights into dynamic gene regulation across diverse biological contexts.

## Supporting information

**S1 Fig. Related to Fig 3.** Other performance metrics except AUPRC on datasets simulated from HARISSA [12], including Area under Receiver Operating Characteristic (ROC) curves and early precision as defined by [16].
(TIF)

**S2 Fig. Related to Fig 5.** AUROC and EP ratio on datasets simulated from curated networks and BoolODE [16]. The EP ratio equals the early precision value divided by the random baseline, and the value should be greater than one if the performance is better than a random classifier.
(TIF)

**S3 Fig. Dependence of median AUPRC on hyperparameters (1)** $(\alpha, \epsilon)$ **for OTVelo-Corr, and (2)**$(\lambda, r)$ **for OTVelo-Granger on HARISSA simulations.** Brighter is better.
(TIF)

**S4 Fig. Dependence of median AUPRC-based metrics on hyperparameters (1)** $(\alpha, \epsilon)$ **for OTVelo-Corr, and (2)**$(\lambda, r)$ **for OTVelo-Granger on Curated models.** Brighter color is better.
(TIF)

**S5 Fig. AUPRC given signed prediction when cross validation is performed to find** $(r, \lambda)$**.** The approach with default parameters $r = 0.5, \lambda = 1$ is labeled 'OTVelo-Granger', and 'OTVelo-CV' indicates a strategy of picking $(r, \lambda)$ via a 5-fold cross validation over a grid $r \in \{0, 0.5, 1.0\}$ and $\lambda \in \{0.1, 0.4, 0.7, 1.0, 1.3, 1.6\}$, the same grid as in Figs S3 Fig and S4 Fig.
(TIF)

**S6 Fig. Dependence of velocity field on** $(\alpha, \epsilon)$ **illustrated on one instance of FN8 dataset simulated by HARISSA.** Top: velocity field with $\alpha = 0.5$ and different $\epsilon$. Lower values of $\epsilon$ gives velocity fields of good smoothness but can take significant computational time or even fail to converge when $\epsilon < 0.001$, while bigger value of $\epsilon$ results in non-smooth velocity field. Bottom: velocity field with fixed $\epsilon = 0.01$ but different $\alpha$. $\alpha = 0$ indicates that one only uses OT cost as in Eq (1), while $\alpha = 1$ indicates pure Gromov–Wasserstein OT that penalizes the change in global structure and is less fine-grained. We show that Gromov–Wasserstein appears to smooth out the velocity field too much while OT can be too restrictive, hence a linear combination of both ($\alpha = 0.5$) can yield a smoother velocity field compared to $\alpha = 0$ or $\alpha = 1$.
(TIF)

**S7 Fig. True network and results from different approaches, illustrated on one instance of FN8 dataset simulated by HARISSA.** Top: resulting graphs from different approaches. Bottom: weight matrices used to construct the graphs. The correlation approach has default parameter $(\alpha, \epsilon) = (0.5, 0.01)$, while the regression approaches all have $\lambda = 1$, and $r = 0, 0.5, 1.0$ respectively. While the correlation was able to capture most of the structure, the two approaches with $l_1$ regression were able to further reduce the density of graph.
(TIF)

**S8 Fig. Related to Fig 10.** Results of HARISSA, CARDAMOM, GENIE3, and SINCERITIES on scGEM dataset [34], with identical layout as Fig 10. GENIE3 does not identify the type of regulation and all edges are visualized in blue.
(TIF)

**S9 Fig. Related to Fig 12.** First two principal components of mouse data according to time and the velocity field identified by optimal transport.
(TIF)

**S10 Fig. Additional AUPRC or AUPRC ratio on predicting undirected and unsigned edges for simulated datasets.** Both ground truth and prediction are symmetrized with edge weights determined by the maximum of absolute value in either directions.
(TIF)

## Acknowledgments

W.Z. would like to thank James Kentro and Tuan Pham for sharing data and background knowledge about biological processes in *drosophila*.

## Author contributions

**Conceptualization:** Wenjun Zhao, Björn Sandstede, Ritambhara Singh.

**Data curation:** Wenjun Zhao.

**Formal analysis:** Wenjun Zhao, Björn Sandstede.

**Funding acquisition:** Björn Sandstede.

**Investigation:** Wenjun Zhao, Björn Sandstede, Ritambhara Singh.

**Methodology:** Wenjun Zhao, Björn Sandstede, Ritambhara Singh.

**Project administration:** Björn Sandstede, Ritambhara Singh.

**Resources:** Björn Sandstede, Ritambhara Singh.

**Software:** Wenjun Zhao.

**Supervision:** Björn Sandstede, Ritambhara Singh.

**Validation:** Wenjun Zhao, Erica Larschan.

**Visualization:** Wenjun Zhao.

**Writing – original draft:** Wenjun Zhao.

**Writing – review & editing:** Wenjun Zhao, Erica Larschan, Björn Sandstede, Ritambhara Singh.

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
