## [Decision Letter · Decision Letter 0]

24 Dec 2024

PCOMPBIOL-D-24-01544

Optimal transport reveals dynamic gene regulatory networks via gene velocity estimation

PLOS Computational Biology

Dear Dr. Zhao,

Thank you for submitting your manuscript to PLOS Computational Biology. After careful consideration, we feel that it has merit but does not fully meet PLOS Computational Biology's publication criteria as it currently stands. Therefore, we invite you to submit a revised version of the manuscript that addresses the points raised during the review process.

Please submit your revised manuscript within 60 days Feb 23 2025 11:59PM. If you will need more time than this to complete your revisions, please reply to this message or contact the journal office at ploscompbiol@plos.org. Please include the following items when submitting your revised manuscript:

We look forward to receiving your revised manuscript.

Kind regards,

Xiuwei Zhang

Guest Editor

PLOS Computational Biology

Shihua Zhang

Section Editor

PLOS Computational Biology

**Journal Requirements:**

3) We notice that your supplementary Figures are included in the manuscript file. Please remove them and upload them with the file type 'Supporting Information'. Please ensure that each Supporting Information file has a legend listed in the manuscript after the references list.

4) Please ensure that the funders and grant numbers match between the Financial Disclosure field and the Funding Information tab in your submission form. Note that the funders must be provided in the same order in both places as well. State the initials, alongside each funding source, of each author to receive each grant. For example: "This work was supported by the National Institutes of Health (####### to AM; ###### to CJ) and the National Science Foundation (###### to AM)." State what role the funders took in the study. If the funders had no role in your study, please state: "The funders had no role in study design, data collection and analysis, decision to publish, or preparation of the manuscript.".

**Reviewers' comments:**

Reviewer's Responses to Questions

**Comments to the Authors:**

Reviewer #1: The manuscript titled "Optimal transport reveals dynanic gene regulatory networks via gene velocity estirratiorl" presents an innovative approach to infer dynamic gene regulatory networks (GRNs) using optimal transport (OT) and gene velocity estimation from time-stamped single-cell RNA sequencing data. The method, OTVelo, integrates optimal transport theory with gene velocity estimates to model gene-to-gene interactions across time points. The authors successfully demonstrate the efficacy and accuracy of OTVelo on simulated, curated, and experimental datasets, offering valuable insights into gene regulatory dynamics during processes such as cell differentiation and reprogramming. The manuscript is well-written and the results are clearly presented, but there are several areas that could benefit from further clarification or elaboration. Below are some revision suggestions that may help improve the manuscript.

1. To improve the manuscript, it is suggested that the quality of the figures be optimized, as some of them appear unclear or low-resolution.

2. The authors should carefully verify the accuracy of the results presented in Figure 7, particularly the AUPR scores, which are below 0.64. These low scores may raise concerns regarding the model's performance, and ensuring the accuracy of the data could enhance the reliability of the conclusions drawn from the results.

3. It is suggested that the comparison of methods could be expanded. Including at least SCENIC, a widely used approach in gene regulatory network inference, would make the comparison more persuasive. This addition would provide valuable insights into the strengths and weaknesses of both methods and help highlight the advantages of the proposed approach.

4. It is suggested that the authors elaborate on the impact of different parameters (e.g., time lag, number of time points) on the performance of the OTVelo framework, potentially through sensitivity analyses or additional experiments.

5. It is suggested to provide a detailed user manual for the software available on GitHub, including instructions on how to run the program and explanations of the parameters. This would help users better understand the software and facilitate its application in different research settings.

Reviewer #2: This paper introduces OTVelo, a tool that uses fused Gromov Wasserstein optimal to estimate gene dynamics from time stamped scRNA-seq data and infer GRN edges using a correlation-based approach and a Granger causality approach. The manuscript is well written with clear description of method details. Open-source codes are provided with easy-to-follow instructions. OTVelo is compared with several other methods on both synthetic datasets and real datasets. My main concern is a relative lack of comparison and clarification on performance on real datasets, in addition to some minor questions about the setup of the method. Overall, I believe this is a novel approach with promising results and is worth publishing if the performance evaluation on real datasets could be improved.

Specific points:

1. It would be helpful to explore the stability of the inferred GRN with respect to the marginal distributions, which are set to uniform distributions in the current work. Some growth/death is expected in multi-time point scRNA-seq data as the time points are often relatively distant. For example, WaddingtonOT uses additional information to estimate a “growth rate”.

2. Could you also demonstrate how using FGW would improve the trajectory constructed using just Wasserstein as is done in WaddingtonOT? It seems counter-intuitive to me as the GW part would preserve the pairwise distances which might lead to missing branching structures.

3. Related to the point above, the quality of the inferred GRN highly depends on the accuracy of the reconstructed trajectory. More evaluations of the trajectory part would be helpful. For example, is it sensitive to choice of parameters including the weights in FGW and the strength of entropy regularization.

4. The better performance of OTVelo on simulated datasets is well demonstrated. However, the comparison to other methods on real data is rather insufficient. I suggest: (1) utilizing the real dataset benchmarks in BEELINE and in a more recent comparison paper: Stock, Marco, et al. "Topological benchmarking of algorithms to infer gene regulatory networks from single-cell RNA-seq data." Bioinformatics (2024): btae267.; (2) include more related methods, such as GRNBoost2; (3) if possible, also evaluate the performance on the task of find undirected edges and compare to methods like PIDC. For (3), I am fine if the authors find undirected edges irrelevant to the focus of the current work. In addition, it would be helpful to elaborate more on the evaluation results. For example, OTVelo outperforms SINCERITIES on signed AUPRC and underperforms SINCERITIES on other metrics including signed AUROC. In what application scenario is signed AUPRC more important than signed AUROC?

5. There is another work that applies OT to GRN inference, though with different approaches. It would be helpful to also survey this method in the introduction:

Lamoline, François, et al. "Gene regulatory network inference from single-cell data using optimal transport." bioRxiv (2024): 2024-05.

Reviewer #3: Inferring gene regulatory networks from gene expression data is an important and challenging problem for the systems biology community. The authors proposed OTVelo, a method for inferring gene regulatory relationships using time-stamped single-cell transcriptome data. OTVelo combines optimal transport and Granger causality, and also uses temporal trajectory to infer gene velocity, which is a meaningful and novel attempt. OTVelo shows comparable or slightly better performance than existing algorithms on several simulated and experimental datasets. Yet the authors also showed that the overall performance of all algorithms is not good enough. GRN inference remains a challenging task and deserves further efforts. To improve this paper, further discussion and tests may be needed.

1. According to Fig. 2 and Fig. 3, it seems that in cases where the network is more complex, the advantages of OTVelo would be more obvious. Can the authors further provide rationales behind such a phenomenon?

2. In Section 3.3.1, Fig. 6, and Fig. S8, the accuracy of gene regulatory networks derived by different algorithms should be quantitatively compared using ground-truth. The authors may quantify how well the regulatory relationships identified by the algorithm match the previously reported relationships. For example, identify how well key regulatory genes inferred by algorithms match with transcription factor activity in corresponding ATAC-Seq or ChIP-Seq data.

3. In Fig. 7, the overall performance of all algorithms is poor. It may not be necessary to overemphasize OTVelo's slightly higher AUPRC value in line 428. The authors may want to objectively discuss why GRN inference is challenging on these experimental data.

4. OTVelo appears to be limited to only some specific datasets. Experimentally derived single-cell transcriptome data not only contain cells in different cell cycles, but also different cell types. This may limit the scope of OTVelo. Can appropriate pre-processing, such as removing cell cycle effects and performing cell classification, help improve OTVelo?

5. PCA is a linear dimension reduction method, which works well when the gene dimension is not high. However, when the genetic dimension is increased and the main component is less significant, PCA will have limited performance. For example, in Fig. 9, PCA dimension reduction cannot reflect cell trajectories over time. Non-linear dimensionality reduction methods such as t-SNE or UMAP are recommended.

6. The authors emphasizes that OTVelo can infer gene velocity and reflect cell velocity on a dimension reduction map. Can the authors compare OTVelo with RNA velocity or methods alike on experimental data?

**Have the authors made all data and (if applicable) computational code underlying the findings in their manuscript fully available?**

Reviewer #1: Yes

Reviewer #2: Yes

Reviewer #3: None

**Figure resubmission:**
---

## [Decision Letter · Decision Letter 1]

10 Apr 2025

Dear Zhao,

We are pleased to inform you that your manuscript 'Optimal transport reveals dynamic gene regulatory networks via gene velocity estimation' has been provisionally accepted for publication in PLOS Computational Biology.

Best regards,

Xiuwei Zhang

Guest Editor

PLOS Computational Biology

Shihua Zhang

Section Editor

PLOS Computational Biology

Reviewer's Responses to Questions

**Comments to the Authors:**

Reviewer #1: There is no further question.

Reviewer #2: The authors have properly addressed all my comments.

Reviewer #3: The authors have addressed my previous comments. One minor issue: Line 482-483 should be described more accurately.

**Have the authors made all data and (if applicable) computational code underlying the findings in their manuscript fully available?**

Reviewer #1: Yes

Reviewer #2: Yes

Reviewer #3: None

---

## [Editor Report · Acceptance letter]

PCOMPBIOL-D-24-01544R1

Optimal transport reveals dynamic gene regulatory networks via gene velocity estimation

Dear Dr Zhao,

I am pleased to inform you that your manuscript has been formally accepted for publication in PLOS Computational Biology. Your manuscript is now with our production department and you will be notified of the publication date in due course.

With kind regards,

Zsofia Freund
